# Intensification and deepening of the Arabian Sea Oxygen Minimum Zone in response to increase in Indian monsoon wind intensity

Zouhair Lachkar[1], Marina Levy[2], and Shafer Smith[1,3]

[1]The Center for Prototype Climate Modeling, New York University in Abu Dhabi, Abu Dhabi, UAE.
[2]Sorbonne Université (UPMC, Paris 6/CNRS/IRD/MNHN), LOCEAN-IPSL, Paris, France.
[3]Courant Institute of Mathematical Sciences, New York University, New York, USA.

*Correspondence to:* Z. Lachkar (zouhair.lachkar@nyu.edu)

**Abstract.** The decline in oxygen supply to the ocean associated with global warming is expected to expand oxygen minimum zones (OMZs). This global trend can be attenuated or amplified by regional processes. In the Arabian Sea, the World's thickest OMZ is highly vulnerable to changes in the Indian monsoon wind. Evidence from paleo records and future climate projections indicate strong variations of the Indian monsoon wind intensity over climatic timescales. Yet, the response of the OMZ to these wind changes remains poorly understood and its amplitude and timescale unexplored. Here, we investigate the impacts of perturbations in Indian monsoon wind intensity (from -50% to +50%) on the size and intensity of the Arabian Sea OMZ, and examine the biogeochemical and ecological implications of these changes. To this end, we conducted a series of eddy-resolving simulations of the Arabian Sea using the Regional Oceanic Modeling System (ROMS) coupled to a nitrogen based Nutrient-Phytoplankton-Zooplankton-Detritus (NPZD) ecosystem model that includes a representation of the $O_2$ cycle. We show that the Arabian Sea productivity increases and its OMZ expands and deepens in response to monsoon wind intensification. These responses are dominated by the perturbation of the summer monsoon wind, whereas the changes in the winter monsoon wind play a secondary role. While the productivity responds quickly and nearly linearly to wind increase (i.e., on a timescale of years), the OMZ response is much slower (i.e., a timescale of decades). Our analysis reveals that the OMZ expansion at depth is driven by increased oxygen biological consumption, whereas its surface weakening is induced by increased ventilation. The enhanced ventilation favors episodic intrusions of oxic waters in the lower epipelagic zone (100-200m) of the western and central Arabian Sea, leading to intermittent expansions of marine habitats and a more frequent alternation of hypoxic and oxic conditions there. The increased productivity and deepening of the OMZ also lead to a strong intensification of denitrification at depth, resulting in a substantial amplification of fixed nitrogen depletion in the Arabian Sea. We conclude that changes in the Indian monsoon can affect, on longer timescales, the large-scale biogeochemical cycles of nitrogen and carbon, with a positive feedback on climate change in the case of stronger winds. Additional potential changes in large-scale ocean ventilation and stratification may affect the sensitivity of the Arabian Sea OMZ to monsoon intensification.

## 1 Introduction

The combination of strong organic matter decomposition and poor ventilation explains the presence of large Oxygen Minimum Zones (OMZs) in the intermediate ocean of the eastern tropical Pacific and Atlantic Oceans as well as in the northern Indian

Ocean. At low oxygen concentrations, hypoxia-sensitive marine species are subject to varying environmental stresses that can affect their growth and reproductive success and ultimately cause their death (Vaquer-Sunyer and Duarte, 2008). Near complete oxygen depletion, suboxic conditions favor anaerobic remineralization of organic matter via denitrification: a process through which nitrate is used as an alternate oxidant. This not only depletes the oceanic inventory of bioavailable nitrogen that is essential for phytoplankton growth, but also releases $N_2O$, a major greenhouse gas that also contributes to stratospheric ozone depletion (Codispoti et al., 2001). Thus, the OMZs not only shape marine ecosystem habitats, but also impact and regulate climate.

Dissolved $O_2$ is predicted to decline in the future in response to upper ocean warming and increased stratification, which may result in the expansion of OMZs (Bopp et al., 2002; Keeling et al., 2010). These changes will have deleterious impacts on marine habitats and may lead to disruption of key biogeochemical cycles (Keeling et al., 2010; Gruber, 2011; Doney et al., 2012). Observational evidence suggests that the oceanic oxygen inventory has already been declining over the recent decades (Helm et al., 2011; Schmidtko et al., 2017) and that OMZs have expanded in several locations (Stramma et al., 2008, 2012). Global model projections show consistently declining oxygen inventories that are commensurate with the warming anomaly, but disagree on the future evolution of OMZs (e.g. Cocco et al., 2013). This may result in part from the misrepresentation of the dynamics of OMZs in global models (Bopp et al., 2013; Cocco et al., 2013; Cabré et al., 2015; Long et al., 2016) and the sensitivity of OMZs to regional climate perturbations that are poorly represented in global simulations.

An example of such perturbations is regional wind changes (e.g. Deutsch et al., 2014). Located nearby major coastal upwelling systems, the OMZs are indeed highly vulnerable to changes in alongshore winds. In most Eastern Boundary Upwelling Systems as well as in the western Arabian Sea, upwelling-favorable wind intensification has been shown to occur under warming scenarios (Wang et al., 2015; deCastro et al., 2016). Bakun (1990) has linked this to increased land-sea thermal gradient under warmer climates, strengthening the land-sea pressure gradient, and hence alongshore winds. Yet, previous observational and model-based studies show that the amplitude of such upwelling intensification strongly varies from one system to another (McGregor et al., 2007; Narayan et al., 2010; García-Reyes and Largier, 2010; Gutiérrez et al., 2011; Barton et al., 2013; Sydeman et al., 2014; Varela et al., 2015).

In the Arabian Sea, the summer monsoon southwesterly winds drive strong upwelling off the coasts of Oman and Somalia, giving rise to one of most productive coastal upwelling ecosystems in the world (Ryther and Menzel, 1965). This high productivity in conjunction with a sluggish circulation sustains the World's thickest OMZ, responsible for up to 40% of global pelagic denitrification despite occupying less than 2% of the World Ocean area (Bange et al., 2005). Future climate projections suggest that Indian summer monsoon may intensify in the future under a warmer climate (e.g. Wang et al., 2013; Sandeep and Ajayamohan, 2015; Praveen et al., 2016; deCastro et al., 2016). Praveen et al. (2016) found that the future upwelling intensification will affect essentially the coast of Oman. Using an ensemble of global and regional model simulations for the 21st century, deCastro et al. (2016) show a strengthening of the Somali coastal upwelling that increases with latitude. Furthermore, these authors show that the intensification of upwelling in the western Arabian Sea is even higher than what is predicted for the Eastern Boundary Upwelling Systems. This study also finds an intensification of land-sea thermal and pressure gradients that is consistent with the Bakun (1990) hypothesis. Other studies reveal that summer monsoon wind intensification has already

been observed recently (Goes et al., 2005; Wang et al., 2013). Goes et al. (2005) reported an increase in upwelling favorable winds off Somalia over the period between 1997 and 2005 and have shown an associated increase in productivity over the same period. Wang et al. (2013) found a global intensification of the Northern Hemisphere summer monsoon since the late 1970s and attributed these trends to the interaction of mega-El Niño/Southern Oscillation and the Atlantic Multidecadal Oscillation, in addition to hemispherical asymmetric global warming.

While there is still no consensus on the magnitude and the drivers of the recent and future Indian monsoon wind changes, evidence from paleoclimate records overwhelmingly suggests a strong link between northern hemisphere temperatures and the Indian monsoon wind intensity on timescales ranging from decades to thousands of years (e.g. Altabet et al., 2002; Schulz et al., 1998; Gupta et al., 2003; Ivanochko et al., 2005). Generally, enhanced Indian summer monsoon intensity was recorded during northern high latitudes warm periods (e.g., Dansgaard-Oeschger events) while reduced summer monsoon intensity was found to correspond to cold periods (e.g., Heinrich events) (Schulz et al., 1998). Ivanochko et al. (2005) has established a relationship between millennial-scale oscillations in summer monsoon intensity and the position of the Intertropical Convergence Zone (ITCZ), where the ITCZ moves northward during warm periods (interstadials) and southward during cold periods (stadials). This relationship between the northern high-latitude climate and the Indian monsoon seen from a multitude of proxies during the last glacial period was also found valid during the Holocene (Gupta et al., 2003). This includes the most recent climate changes from the Medieval Warm Period and the Little Ice Age. The analysis of paleo proxy records also suggests a strong coupling between northern climate excursions and changes in productivity and denitrification in the Arabian Sea at different timescales (e.g. Altabet et al., 1995, 1999, 2002; Gupta et al., 2003; Singh et al., 2011). For instance, Altabet et al. (1999) found that denitrification was greatest during interglacial periods and was probably not active during most glacial phases. Altabet et al. (2002) found a strong correspondence between changes in the productivity and denitrification of the Arabian Sea and century-scale Dansgaard-Oeschger events during the last glacial period. These studies show that denitrification increases during warm periods concurrent with the summer monsoon and productivity intensification and decreases during cold phases. Other studies have highlighted that the fluctuations in denitrification and the intensity of the OMZ can also be driven by changes in winter monsoon wind intensity (Reichart et al., 2004; Klöcker and Henrich, 2006).

Despite these previous studies, the amplitude of the OMZ sensitivity to potential wind changes remains largely uncertain. Indeed, whether in the context of past climate fluctuations or under future climate change, the response of OMZ to upwelling-favorable wind intensification is difficult to predict as such a perturbation may increase both oxygen supply through enhanced ventilation and oxygen demand via increased biological productivity, thus leading to an uncertain net effect. In the Arabian Sea, the picture is made even more complicated by the seasonal reversal of winds and the potential importance of changes in winter monsoon mixing. Additionally, the large denitrification fluxes in the Arabian Sea combined with the potential feedback of denitrification on biological productivity, and hence on oxygen consumption, further add to the intricacy of the problem. Finally, the question of the OMZ response timescales is essential but remains unanswered. Here we address these questions and explore the mechanisms by which the Arabian Sea ecosystem responds to monsoon wind changes using a regional eddy-resolving model. We examine how idealized changes in summer and winter monsoon wind intensity affect the productivity and the volumes of hypoxic and suboxic water in the Arabian Sea and explore the biogeochemical and ecological implications

of these changes. We show that the productivity increases on a timescale of years while the OMZ expands and deepens on a timescale of decades. This response is essentially driven by summer monsoon wind intensification and results from an enhanced biological consumption of oxygen opposed by increased ventilation near the surface. The enhanced upper ocean ventilation leads to intermittent expansions of habitats in the epipelagic zone wile the OMZ intensification at depth increases denitrification, thus amplifying the depletion of bioavailable nitrogen in the Arabian Sea. We conclude that changes in the Indian monsoon can affect the large-scale nitrogen marine budget on decadal to centennial timescales, with a positive feedback on warming in the case of stronger winds.

## 2   Methods

### 2.1   Models

We use the Regional Oceanic Modeling System (ROMS)-Agrif (documented at http://www.romsagrif.org/) configured for the Arabian Sea region. ROMS solves the primitive equations and has free surface and generalized terrain-following vertical coordinates (Shchepetkin and McWilliams, 2005). Advection is represented using a rotated-split third-order upstream biased operator designed to limit dispersive errors and preserve low diffusion (Marchesiello et al., 2009). The subgrid vertical mixing is represented using the nonlocal K-Profile Parameterization (KPP) scheme (Large et al., 1994). Numerical diffusivity allows the dissipation of small-scale noise as no additional background lateral diffusivity is used. The ecological-biogeochemical model is a nitrogen-based nutrient-phytoplankton-zooplankton-detritus (NPZD) model (Gruber et al., 2006; Lachkar et al., 2016). It is based on a system of ordinary differential equations describing the time evolution of the following tracers: nitrate ($NO_3^-$), ammonium ($NH_4^+$), one class of phytoplankton, one class of zooplankton, two classes of detritus and a dynamic chlorophyll-to-carbon ratio. The two pools of detritus represent respectively large organic matter particles that sink fast ($10\,\mathrm{m\,d^{-1}}$) and small particles that sink at slower speed ($1\mathrm{m\,d^{-1}}$). The small particles can coagulate with phytoplankton to form fast sinking large detritus. The sinking of the particles is represented explicitly in the model, thus allowing all tracers to be advected laterally including in the euphotic zone. When sinking particles reach the seafloor, they are remineralized back into ammonium at a much slower rate ($0.003\,\mathrm{d^{-1}}$) than in the water column ($0.03\,\mathrm{d^{-1}}$ for small detritus and $0.01\,\mathrm{d^{-1}}$ for large detritus). Finally, the model is coupled to a module that describes the biological sources and sinks of oxygen following Lachkar et al. (2016). Furthermore, at very low oxygen concentrations ($O_2 < 4\,\mathrm{mmol/m3}$), nitrification and aerobic remineralization are shut down and denitrification is set to consume nitrate instead of oxygen. Finally, benthic denitrification is parameterized following Middelburg et al. (1996) (see further details of the model in Lachkar et al. (2016)).

### 2.2   Experimental design

The model domain extends in latitude from 5°S to 30°N and in longitude from 34°E to 78°E , thus encompassing the whole Arabian Sea OMZ as well as the key dynamical features of the circulation of the region such as the Somali and Omani upwelling systems and the Great Whirl. The model horizontal grid has a resolution of 1/12° with an average grid spacing that is about

5 to 20 times smaller than the local first baroclinic Rossby deformation radius (Chelton et al., 1998). This permits an explicit representation of a large fraction of the region eddies. The vertical grid consists of 32 sigma layers with an improved resolution near the surface. The seafloor topography is derived from the ETOPO2 dataset supplied by the National Geophysical Data Center (Smith and Sandwell, 1997).

The model is forced with a monthly climatology. The absence of interannual variability in the atmospheric forcing enables us to quantify the role of internal variability associated with mesoscale eddies. The temperature, salinity and currents are initialized and laterally forced using the Simple Ocean Data Assimilation (SODA) ocean reanalysis. Oxygen and nitrate initial and boundary conditions are derived from the World Ocean Atlas (2009) dataset. The atmospheric boundary conditions for heat and freshwater fluxes are based on the Comprehensive Ocean-Atmosphere Data Set (COADS) (da Silva et al., 1994). We

applied a surface restoring to COADS observed surface temperatures and salinities using kinematic flux corrections described in Barnier et al. (1995). The model is forced with wind stress data from the QuikSCAT-derived Scatterometer Climatology of Ocean Winds (SCOW) (Risien and Chelton, 2008).

The model is first spun up for 12 years and then is run for an additional 50 years using 9 different wind stress scenarios. In the control run the wind stress is left unperturbed. In a first set of four perturbed monsoon simulations the wind stress was

uniformly increased or decreased by 20% and 50%, respectively. This amounts to wind speed perturbations of around 10 and 20%, respectively. Two additional runs were conducted where the perturbations of the summer and winter monsoon winds are set to be antagonistic, i.e., a 50% increase (resp. decrease) of the summer monsoon wind stress that is concomitant with a 50% decrease (resp. increase) in the winter monsoon wind stress. Finally, two simulations with gradual increase (resp. decrease) of wind stress at a rate +1% per year are performed.

Although these runs explore different wind perturbation scenarios, they are highly idealized by nature and are not intended to mimic past conditions from paleoclimatic reconstructions or realistic future trajectories, but rather aim at exploring the sensitivity of the Arabian Sea OMZ to monsoon wind intensity changes and improving our understanding of the key mechanisms that control the OMZ response and its timescales. For model evaluation we use the last 10 years of the control run.

## 2.3   Model evaluation

We use available satellite and in-situ observations to assess the model ability to reproduce observed physical and biogeochemical properties in the Arabian Sea domain. To this end, we evaluate the model in terms of surface currents and eddy kinetic energy (EKE), sea surface height (SSH) anomalies, sea surface temperature (SST) and surface chlorophyll-a concentrations. Furthermore, we compare modeled primary production and export fluxes with estimates based on available field data and satellite observations. Finally, we examine the distributions of temperature, salinity, oxygen and nitrate in the upper ocean in the

model and in the World Ocean Atlas (2013) dataset.

The model simulates successfully the surface eddy kinetic energy (EKE) as it reproduces quite accurately the spatial distribution of the observed surface eddy field (Fig. 1). In particular, the observed east-west gradient in EKE is captured by the model. However, the model tends to slightly underestimate the magnitude of the eddy activity in the eastern and northeastern Arabian Sea. This might be due to the resolution of the model that does not permit yet the representation of the full eddy

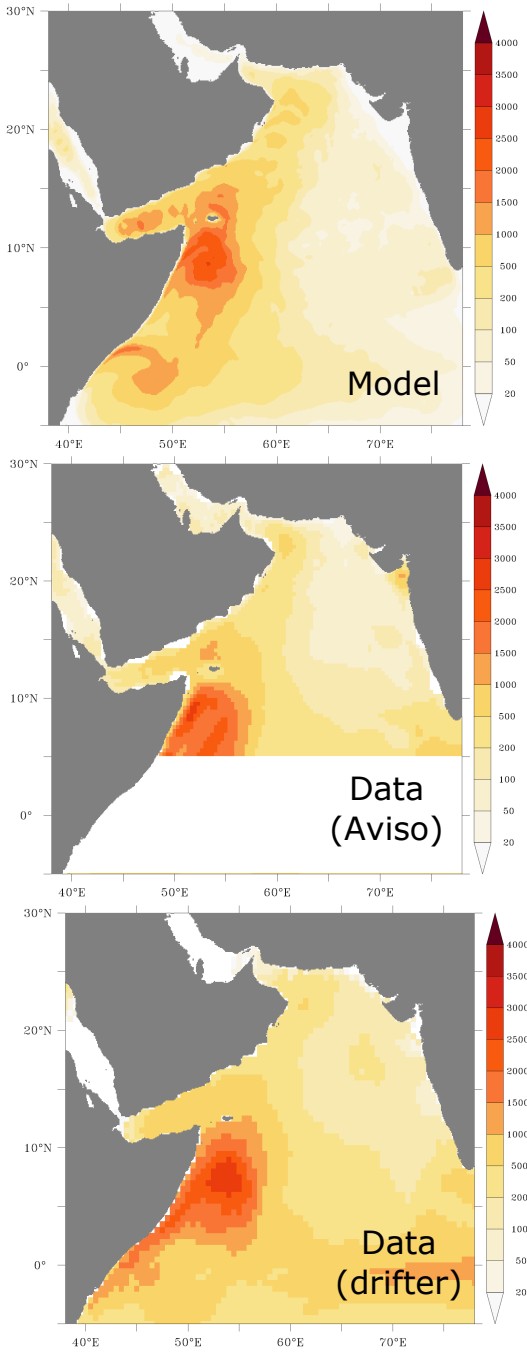

**Figure 1. Surface eddy kinetic energy.** Surface eddy kinetic energy (in $\mathrm{cm}^2\,\mathrm{s}^{-2}$) as simulated in the model (top) and from data based on Aviso satellite altimeter (middle) and surface drifter climatology of Lumpkin and Johnson (2013) (bottom).

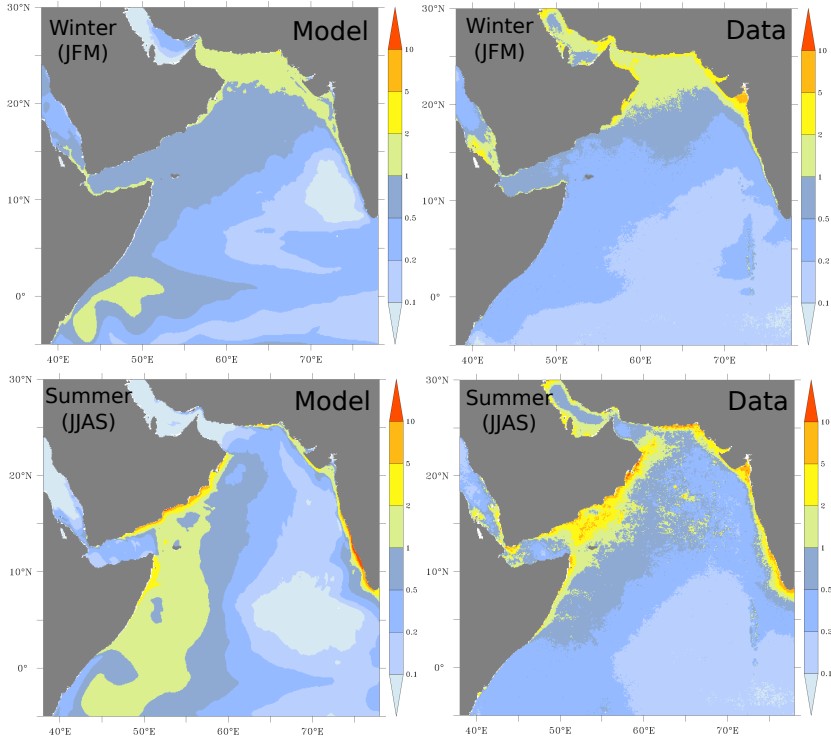

**Figure 2. Surface distribution of chlorophyll-a.** Surface chlorophyll-a concentrations (in mg m$^{-3}$) as simulated in ROMS (left) and from SeaWiFS (right) during winter (top) and summer (bottom) months. The SeaWiFS climatology is computed over the period from 1997 to 2009.

spectrum in this region (Chelton et al., 1998; Lachkar et al., 2016). The model also captures the main patterns of the observed surface circulation both in summer and winter seasons (Appendix A: Supplementary figures, Fig. A1). In particular, the model reproduces the Northeast Monsoon Current and the South Equatorial Countercurrent in winter and the energetic Great Whirl and Southern Gyre in summer (Schott and McCreary, 2001). Furthermore, the strength and the seasonal reversal of the Somali

5 Current are also correctly reproduced in the model. Finally, the model reproduces the observed seasonal sea surface height (SSH) anomalies in both winter and summer seasons (Appendix A: Supplementary figures, Fig. A2). In particular, both the spatial distribution and the magnitude of coastal SSH gradients are accurately captured.

The model captures the main patterns of the observed sea surface temperature (SST) from AVHRR satellite data in winter and summer seasons (Appendix A: Supplementary figures, Fig. A3). In particular, the model reproduces the east-west and north-

10 south temperature gradients across the model domain, as well as the strong surface gradients between the Arabian Sea and its marginal seas (i.e., the Gulf and the Red Sea). Furthermore, the model also captures the cold SST tongue that characterizes summer upwelling off the coasts of Oman and Somalia.

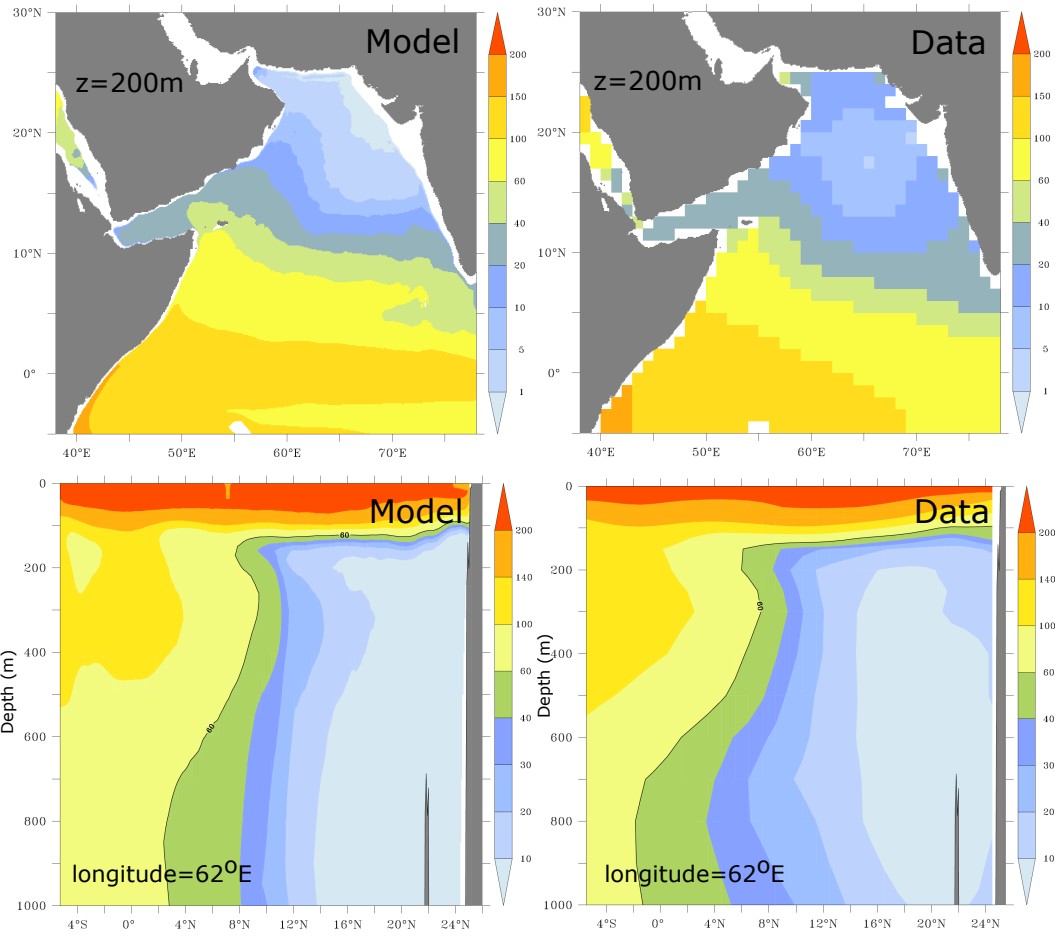

**Figure 3. Horizontal and vertical distributions of oxygen.** Distribution of annual mean oxygen (in mmol m$^{-3}$) (top) at 200 m and (bottom) in the upper 1000 m along 62°E as simulated in ROMS (left) and from World Ocean Atlas (2009) dataset.

However, an examination of the north-south vertical distribution of temperature reveals that the model: (i) tends to underestimate the subsurface water temperature in the northern Arabian Sea particularly in winter and (ii) slightly overestimates the mixed layer depth in both seasons (Appendix A: Supplementary figures, Fig. A4). Similar mismatch with observations characterizes the simulated salinity fields (Fig. A3 and Fig. A4). Although the model generally reproduces the observed surface distribution of salinity, it underestimates the deep penetration of salty surface waters in the northern region of the Arabian Sea. This high-salinity water is characteristic of the Gulf water and its underestimation in our model indicates that the model may underestimate the subduction of the Gulf water into the Arabian Sea. This might be due to model limitations such as the lack of adequate vertical and horizontal resolutions as well as due to potential inconsistencies in atmospheric forcing data over the Gulf region.

The model successfully simulates the high chlorophyll concentrations in the northern and western Arabian associated with the winter and summer blooms, respectively (Fig. 2). The model also reproduces the high chlorophyll concentrations that are observed off the Indian west coast in both seasons. However, the model substantially overestimates the observed chlorophyll concentrations off the Somali coast south of 4°N in both seasons. This is likely due to our nitrogen based NPZD model that does not account for potential limitation of productivity by silicic acid and iron in this region as documented by Koné et al. (2009). Besides this model deficiency, the large-scale distribution of chlorophyll is fairly well represented in most of the Arabian Sea despite a tendency to underestimate chlorophyll in the open ocean. This might have to do with the model's lack of small phytoplankton functional groups that are more adapted to oligotrophic conditions. Overall, the fidelity of the modeled chlorophyll is, however, in line with that simulated by state of the art models of the Indian Ocean and the Arabian Sea (e.g. Resplandy et al., 2012).

We further compare simulated primary production and export fluxes with estimates based on available field data from the US JGOFS Arabian Sea Process Study (Lee et al., 1998). These consist in: i) in-situ measured primary production ii) export fluxes at 100 m estimated using $^{234}$Th removal rates and iii) export fluxes estimated from sediment trap data at 500 m above the seafloor. Fluxes were measured essentially during the year 1995 at 5 sites (M1, M2, M3, M4 and M5) along a transect extending from the Coast of Oman to the central Arabian Sea (Lee et al., 1998). Because of the limited number of in-situ observations of productivity (only 5 measurements at each site), we also use satellite-based productivity estimates obtained from two sensors (SeaWiFS and MODIS) and based on two different algorithms: the Vertically Generalized Production Model (VGPM) (Behrenfeld and Falkowski, 1997) and the Carbon Based Production Model (CBPM) (Westberry et al., 2008). The model correctly simulates the observed decrease in productivity associated with increasing distance to coast (Appendix A: Supplementary figures, Fig. A5). Yet, it substantially underestimates the in-situ based estimates at all sites. Some of this mismatch may be due to the fact that the in-situ estimates are derived from a small number of independent measurements that are all coming from one individual year (1995). Given the importance of both mesoscale and interannual variability, these estimates may therefore not be representative of the long-term climatological conditions simulated by the model. Indeed, a better agreement is obtained between the modeled productivity and estimates based on satellite observations that have a more extensive temporal coverage (Fig. A5). We further contrasted the simulated export fluxes at 100 m to estimates from Lee et al. (1998) at the 5 stations (Fig. A6a). Our modeled export fluxes generally overestimate the $^{234}$Th-based estimates but remain comparable to them in magnitude. Furthermore, the model reproduces quite accurately the observed offshore gradient in export. It is worth highlighting however that similarly to the in-situ measured productivity, these export fluxes are based on 4 independent measurements at each site only, all from the same year (1995). Finally, comparing the modeled export fluxes in the deep ocean (500 m above the seafloor) to sediment trap data at the same sites yields a good agreement between the model and the observations in all stations (Fig. A6b). It is worth noting that these deep export flux estimates can be considered as more robust than those at 100 m as they are based on a larger number of independent measurements (20-40 measurements at each site).

The simulated surface distribution of nitrate is consistent with observations from the World Ocean Atlas dataset in both seasons (Fig. A7). In particular, the high surface concentrations associated with the summer upwelling in the western Arabian Sea

and the winter convection in the northern Arabian Sea are captured by the model. Finally, the simulated oxygen distributions at depth compare generally well with observations (Fig. 3). In particular, the location, the size and the depth of the Arabian Sea oxygen minimum zone (OMZ) (defined here by $O_2 < 60$ mmol m$^{-3}$) are correctly reproduced. However, the model overestimates the intensity of the OMZ in the northern Arabian Sea and off the Indian west coast. This might be driven by the model underestimated eddy activity that may lead to underestimated OMZ ventilation as shown by Lachkar et al. (2016). This might also partly result from the misrepresentation and likely underestimation of the injection of the Gulf waters into the northern Arabian Sea at intermediate depths.

A more quantitative evaluation of model skill is performed using in-situ observations from the World Ocean Database (2013) that we binned into a 0.5° monthly climatology. No spatial interpolation is made and the model and observations are compared at the observation points. The results of this evaluation are summarized graphically using Taylor diagrams (Taylor, 2001) quantifying the similarity between the observations and model in terms of their correlation, the amplitude of their standard deviations and their centered root-mean-square (RMS) difference (Fig. 4). Our statistical analysis shows that the simulated and observed SSH anomalies have similar standard deviations and correlate strongly with correlation coefficients of 0.77 and 0.83 in winter and summer, respectively (Fig. 4a). Similarly, the quantitative comparison of the simulated and observed SST climatologies shows similar standard deviations and small RMS errors as well as very high correlations ($r > 0.92$) in both seasons, confirming the visual comparison conclusions (Fig. 4a). Likewise, the simulated and observed chlorophyll distributions show relatively low RMS errors in both seasons with decent correlations ranging from 0.69 in summer to 0.74 in winter (Fig. 4a). Despite the underestimation of the penetration of the warm and salty surface waters in the northern Arabian Sea at depth, the large scale distributions of temperature and salinity in the upper ocean are captured by the model as indicated by low RMS errors and very high correlations with in-situ observations (r > 0.91) for both temperature and salinity (Fig. 4b). Additionally, the model shows an overall high skill in reproducing the observed large-scale distributions of both nitrate and oxygen, with correlations with the World Ocean Database ranging in the upper 200 m from 0.88 for nitrate in summer months to around 0.93 for oxygen in both seasons (Fig. 4b).

In summary, despite some local biases, the model generally shows reasonable skill in reproducing the large-scale features of the circulation in the Arabian Sea as well as the seasonal dynamics of phytoplankton blooms in the region. More importantly, it reproduces fairly well the location and structure of the Arabian Sea oxygen minimum zone. The discussion of the potential impact of the model limitations on our results will be addressed in detail in the discussion section.

## 3 Results

To explore the sensitivity of the Arabian Sea ecosystem to changes in the intensity of monsoon winds, we consider various scenarios of idealized wind perturbations. A first set of scenarios consists in increasing (resp. decreasing) the wind stress over the whole domain and throughout the year by 20% and 50%, respectively. In a second set of experiments, we increase the wind stress by 50% in summer (resp. winter) and decrease it by 50% in winter (resp. summer). This is to account for perturbation scenarios where summer and winter monsoon winds evolve in opposite directions as suggested by multiple paleo

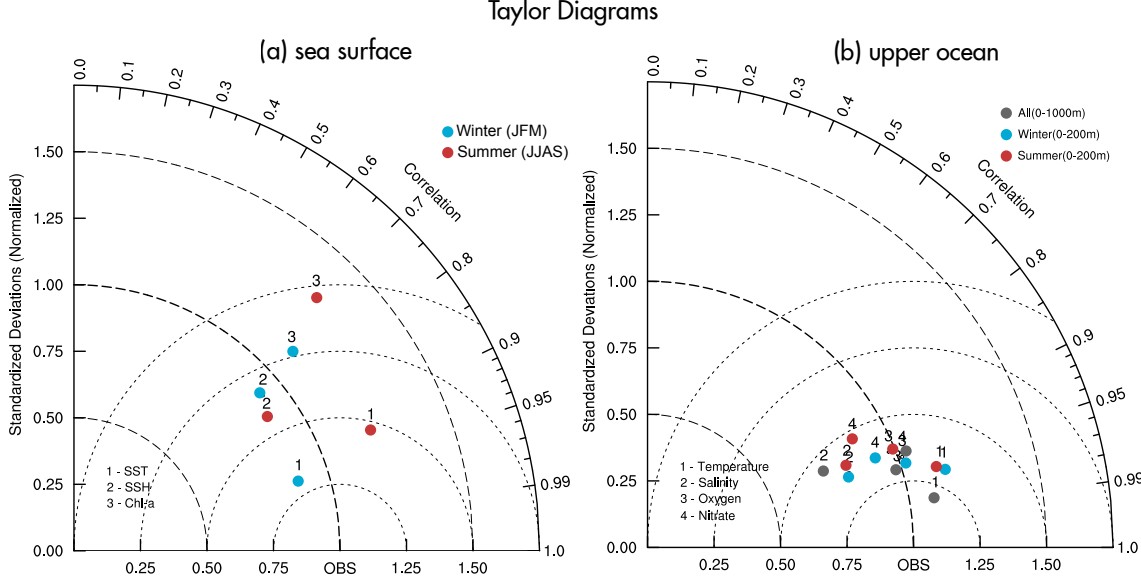

**Figure 4. Taylor diagram displaying statistical comparison of modeled and observed fields.** Taylor (2001) diagram of simulated (a) sea surface temperature (data points labeled "1"), sea surface height anomalies (data points labeled "2") and surface chlorophyll-a (data points labeled "3") in winter (blue) and summer (red) months and (b) temperature (data points labeled "1"), salinity (data points labeled "2"), oxygen (data points labeled "3") and nitrate (data points labeled "4") in the upper 1000 m and in the top 200 m in winter (blue) and summer (red) months. The reference point of the Taylor diagram corresponds to (a) SeaWiFS observations for chlorophyll, AVHRR data for surface temperature and AVISO climatology for sea surface height anomalies and (b) World Ocean Database 2013 for all four variables. The radius (distance to the origin point) represents the modeled standard deviation relative to the standard deviation of the observations. The angle between the model point and the X-axis indicates the correlation coefficient between the model and the observations. Finally, the distance from the reference point to a given modeled field represents that field's centered root mean square (RMS) with respect to observations.

records (e.g. Klöcker and Henrich, 2006) showing intensification of summer monsoon to be concomitant with the weakening of winter monsoon and vice versa. This will also allow us to disentangle the impacts of the changes in each monsoon season and determine their respective contributions to the overall response. The wind perturbations are applied instantly and maintained over 50 years for each scenario (abrupt perturbation). The impact of perturbations entailing gradual changes in the wind

5   intensity will be addressed later in section 4.

### 3.1   Response of NPP

The magnitude of the Arabian Sea net primary production (NPP) response is proportional to the amplitude of the perturbation (Fig. 5). For instance, NPP increases (resp. decreases) by around 20% and 45% to 50% in response to an increase (resp. decrease) in the wind stress of 20% and 50%, respectively. However, when the summer wind intensification (resp. weakening)

10   is concomitant with a decrease (resp. increase) in the winter monsoon winds, the annual mean NPP still shows an increase (resp.

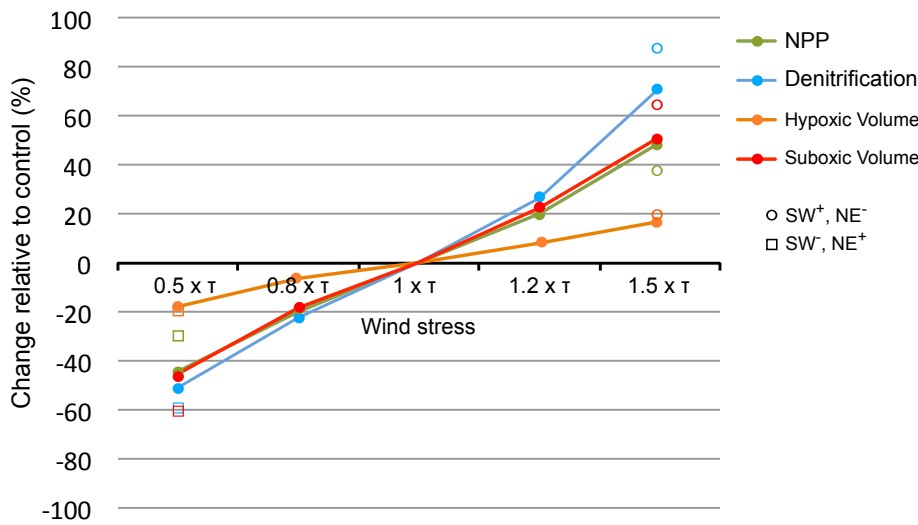

**Figure 5. Biogeochemical response to monsoon wind intensity changes.** Relative changes in response to monsoon wind intensity perturbations in net primary production (green), denitrification (blue), suboxic volume (red) and hypoxic volume (orange). Open circles (resp. squares) indicate the results from the simulation where summer monsoon wind stress is increased (resp. decreased) by 50% and winter monsoon wind stress is decreased (resp. increased) by 50%.

decrease) but of a weaker amplitude. Therefore, we conclude that the NPP response is dominated by the summer southwest monsoon (SWM) perturbation and that the winter northeast monsoon (NEM) wind changes play a minor role. Two factors explain the strong control of the SWM perturbation over the NPP annual mean response. First, the biological production during the SWM dominates the annual production (explains more than 40% of the annual levels while NEM productivity contributes

5   by less than 33%). Second, summer productivity is more sensitive to wind changes as it is directly driven by wind-induced upwelling. In contrast, NEM productivity is driven by wintertime cooling and convection. Hence, NEM wind intensification enhances vertical mixing and surface nutrient concentrations, but also deepens the mixed layer, thus potentially increasing light limitation. Indeed, winter turbulent mixed layer deepens in the northern Arabian Sea by up to 20-25m and penetrates below the euphotic zone (1% light depth) at 65-70m. This increases the average exposure of phytoplankton to light-limited

10  conditions, thus potentially limiting the net growth rate (i.e.., gross photosynthetic rate minus loss terms due to mortality, grazing, sinking and respiration) over the water column, and hence reducing the potential biomass and productivity (Franks, 2015). This may explain the more limited increase in winter productivity (+38% increase in response to 50% increase in wind stress) in comparison to summer productivity (+52% increase in response to 50% increase in wind stress). Finally, the response of NPP to changes in Indian monsoon wind intensity develops over a relatively short timescale (Fig. 6 and Fig. A8). Indeed,

15  the NPP reaches a quasi steady state with limited internal variability within 3 to 5 years from the start of the perturbation.

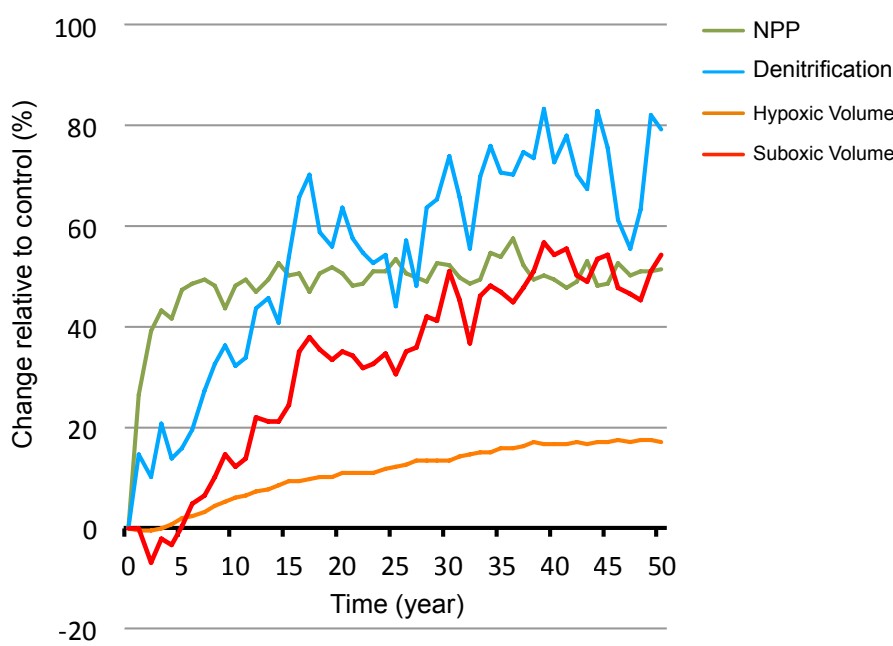

**Figure 6. Response to monsoon wind increase as a function of time.** Response to wind stress increase (+50%) of NPP, denitrification and suboxic and hypoxic volumes as a function of time.

### 3.2 Response of OMZ and denitrification

Under monsoon wind intensification (resp. weakening), the OMZ (defined here as hypoxic water with $O_2 < 60$ mmol m$^{-3}$) expands (resp. contracts) (Fig. 5). The OMZ response is much slower and of a weaker amplitude than the NPP response (Fig. 6 and Fig. A8). For instance, it takes several decades for the hypoxic volume to reach a quasi steady state. Furthermore, the OMZ expands (resp. contracts) by a mere 20% in response to a 50% wind stress increase (resp. decrease). Similarly to NPP, the OMZ response is dominated by the summer perturbation (Fig. 5). This can be partially explained by the larger summer productivity and its larger sensitivity to wind changes leading to stronger perturbation of the $O_2$ demand. Additionally, the deepening (by up to 25m) of the wintertime mixed layer that result from NEM intensification enhances the ventilation of the northern and northeastern Arabian Sea, thus compensating the mild increase in $O_2$ consumption that result from enhanced winter productivity. Finally, the response of the suboxic volume (defined here as $O_2 < 4$ mmol m$^{-3}$) is similarly slow (30 to 50 years) but of much larger amplitude (Fig. 5 and Fig. 6). Indeed, the suboxic volume expands by around 50% in response to a 50% increase in the wind stress. The amplitude of this response is even larger (more than 60%) when the summer monsoon intensification occurs concurrently with a weakening of the winter monsoon winds (Fig. 5). However, the response of the suboxic volumes comes with a substantially stronger internal variability (Fig. 6).

As denitrification develops only under suboxic conditions, its response to monsoon changes is modulated by the response of the suboxic volume. Similarly, it shows a slow response (on a timescale of several decades) characterized with strong internal

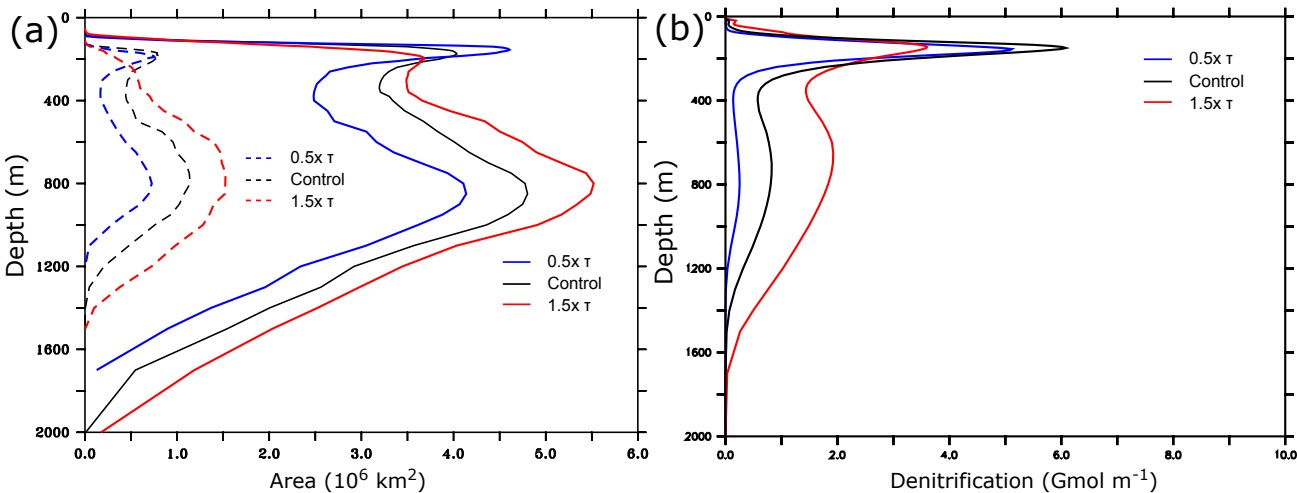

**Figure 7. Changes in the Arabian Sea OMZ and denitrification as a function of depth.** (a) Total area (in $10^6$ km$^2$) occupied by hypoxic (solid lines) and suboxic (dashed lines) waters as a function of depth under different monsoon wind intensities. (b) Arabian Sea integrated annual-mean denitrification (in $10^9$ mol m$^{-1}$) as a function of depth. Note the presence of a weak secondary maximum in domain-integrated denitrification between 500 and 800 m in all simulations. In the control simulation, this can be explained by the fact that at this depth range the area occupied by (potentially denitrifying) suboxic water is largest and weak denitrification is still present although at an order of magnitude weaker rate (on per unit of volume basis) than at 200 m.

variability (Fig. 6). However, the amplitude of denitrification response is generally larger than that of suboxia (Fig. 5). For instance, denitrification increases by more than 85% when summer monsoon intensifies by 50% and winter monsoon weakens by 50%, whereas the suboxic volume increases by only around 60% under the same perturbation. This is consistent with denitrification fluxes depending both on the volume of suboxia as well as on the amplitude of production fluxes. As both NPP
5   and suboxia increase under monsoon intensification, the cumulative effect of their increases lead to a larger denitrification response.

Under increased monsoon winds, hypoxia increases at depth but is reduced in the upper ocean (Fig. 7a). Similarly, the suboxic volume expands at depth and contracts near the surface. Indeed, the peak of low O$_2$ located in the lower epipelagic zone (150-200 m) as seen in Fig. 3 weakens under enhanced monsoon winds, whereas the lower OMZ peak ($<$ 400 m) expands.
10  This results in the intensification of denitrification at depth and its attenuation in the upper ocean (Fig. 7b). These changes involve varying timescales and contrasting levels of internal variability (Fig. 8 and Fig. A8). The compression of the OMZ in the epipelagic zone (0-200 m) is relatively quick (i.e., a timescale of years) and is associated with a strong internal variability. On the other hand, the OMZ expansion is much slower (i.e, a timescale of decades) in the intermediate (i.e., mesopelagic zone 200-1000m) and the deep (i.e., bathypelagic zone >1000m) oceans and is accompanied by a much weaker internal variability.

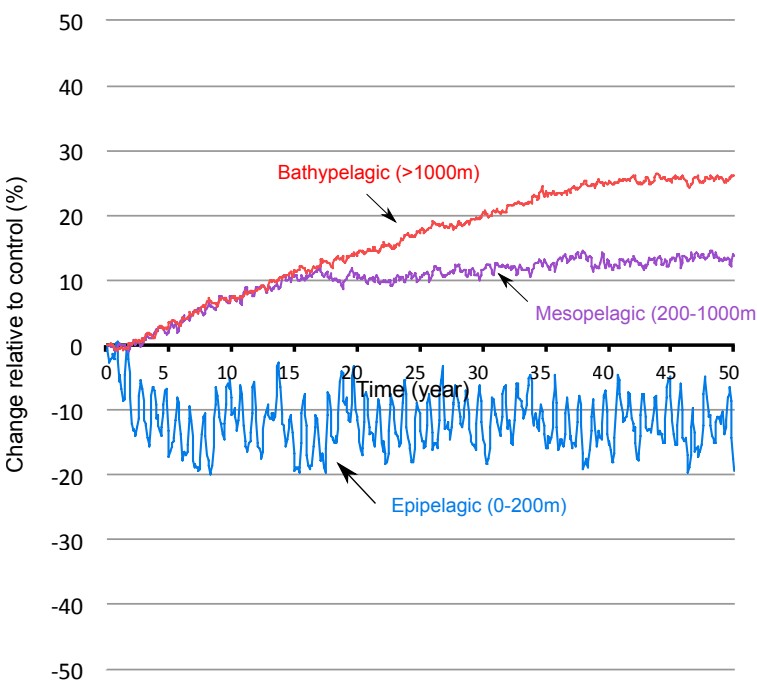

**Figure 8. OMZ response in different vertical layers as a function of time.** Changes in hypoxic volume (in %) under wind stress increase (+50%) in the epipelagic zone (0-200m), the mesopelagic zone (200-1000m) and the bathypelagic zone (>1000m).

## 4 Discussion

### 4.1 Drivers of the OMZ change and its timescale

In order to elucidate the factors driving the OMZ changes, we perform an oxygen budget analysis in the OMZ volume. We first take into account the whole water column and then carry out the analysis separately in three layers: the epipelagic zone 5 (0-200m), the mesopelagic zone (200-1000m) and the bathypelagic zone (>1000m). For simplicity we consider only the two most extreme scenarios where the wind stress is increased or decreased by 50%. For each simulation we subtract the control run from the perturbation run and focus on the anomalies with respect to the non-perturbed case. We quantify the annual oxygen accumulation within the OMZ with respect to the control run and determine the contributions of transport and biology to this (Fig. 9 and Fig. A9).

10 This analysis shows that when considering the whole OMZ, the oxygen accumulation is generally negative (corresponding to a net loss of $O_2$) over the first two decades when monsoon winds are increased (Fig. 9a). However, this term becomes gradually small and oscillates around zero during the last decade of the simulation as the OMZ tends towards a different steady state. This explains the fast increase in the OMZ volume over the first few decades and its quasi stabilization by the end of the simulation. This oxygen loss is driven by an imbalance between increased $O_2$ supply from transport and enhanced $O_2$ consumption due 15 to biology (Fig. 9a). Over the first 20 to 30 years under increased winds, the enhanced supply of oxygen through ventilation

undercompensates the additional sink of oxygen due to biology. This is because the biological response is much quicker than the ventilation response. Indeed, it takes around 30 years for the transport contribution to reach a quasi equilibrium, and only around 3 years for the biological response to reach its steady state. This difference in the response timescale of biology and circulation is responsible for the slow expansion of OMZ. Additionally, most of the internal variability that characterizes the

OMZ oxygen content (and hence its size and intensity) is associated with the ventilation contribution.

The oxygen budget in the upper (0-200m) OMZ reveals that the $O_2$ accumulation term is predominantly positive over the first few years of the simulation, and is driven by a supply of oxygen through transport that exceeds biological consumption (Fig. 9b). The oxygen source and sink reach a quasi balance within a couple of years. This explains the rapid response of the OMZ in the upper ocean in our model. The strong internal variability that characterizes the time evolution of hypoxic volume

is driven by the variability of the transport of oxygen to the OMZ. In the intermediate mesopelagic zone (200-1000m), the slow OMZ response is driven by a much slower response of the circulation driven oxygen supply that only balances the quick biological response with a delay of 2 to 3 decades (Fig. 9c). In the deep bathypelagic zone (>1000m), the response of the OMZ is even slower because the circulation timescale is even longer (Fig. 9d).

## 4.2   Implications and limitations of the study

The oxygen changes induced by monsoon wind perturbation have important ecological and biogeochemical implications. In particular, changes that affect the upper OMZ can have direct effects on marine habitats and may impact the ecosystem community structure. On the other hand, the changes in the OMZ intensity have the potential - via denitrification - to alter the marine nitrogen budget, and hence the efficiency of the biological pump of carbon and climate, on the longer timescales.

### 4.2.1   Implications for habitats

A 50% increase in the monsoon wind stress leads to almost a doubling of the surface eddy kinetic energy in the central and western Arabian Sea (Fig. 10). As mesoscale eddy variability explains most (> 85%) of subsurface oxygen variance in the upper OMZ of the region, this enhanced eddy activity further amplifies the already high $O_2$ variability of the western Arabian Sea, and increases the alternation of hypoxic and oxic conditions there (Fig. 11). This can enhance marine habitat variability, and hence strongly impact local ecosystem community structure. The rich yet fragile west Arabian Sea ecosystem faces already

several local and large-scale stressors such as harmful algal blooms, surface warming and ocean acidification (e.g. de Moel et al., 2009; Gomes et al., 2014; Roxy et al., 2016). Habitat changes associated with wind-driven $O_2$ perturbations in the upper ocean could add to these stressors and lead to profound ecosystem changes. It is worth recalling that as our model is forced with monthly climatology, the increase in habitat variability under intensified monsoon simulated here does not take into account potential changes in high frequency atmospheric forcing. These may further amplify or dampen habitat variability depending

on their potential future evolution.

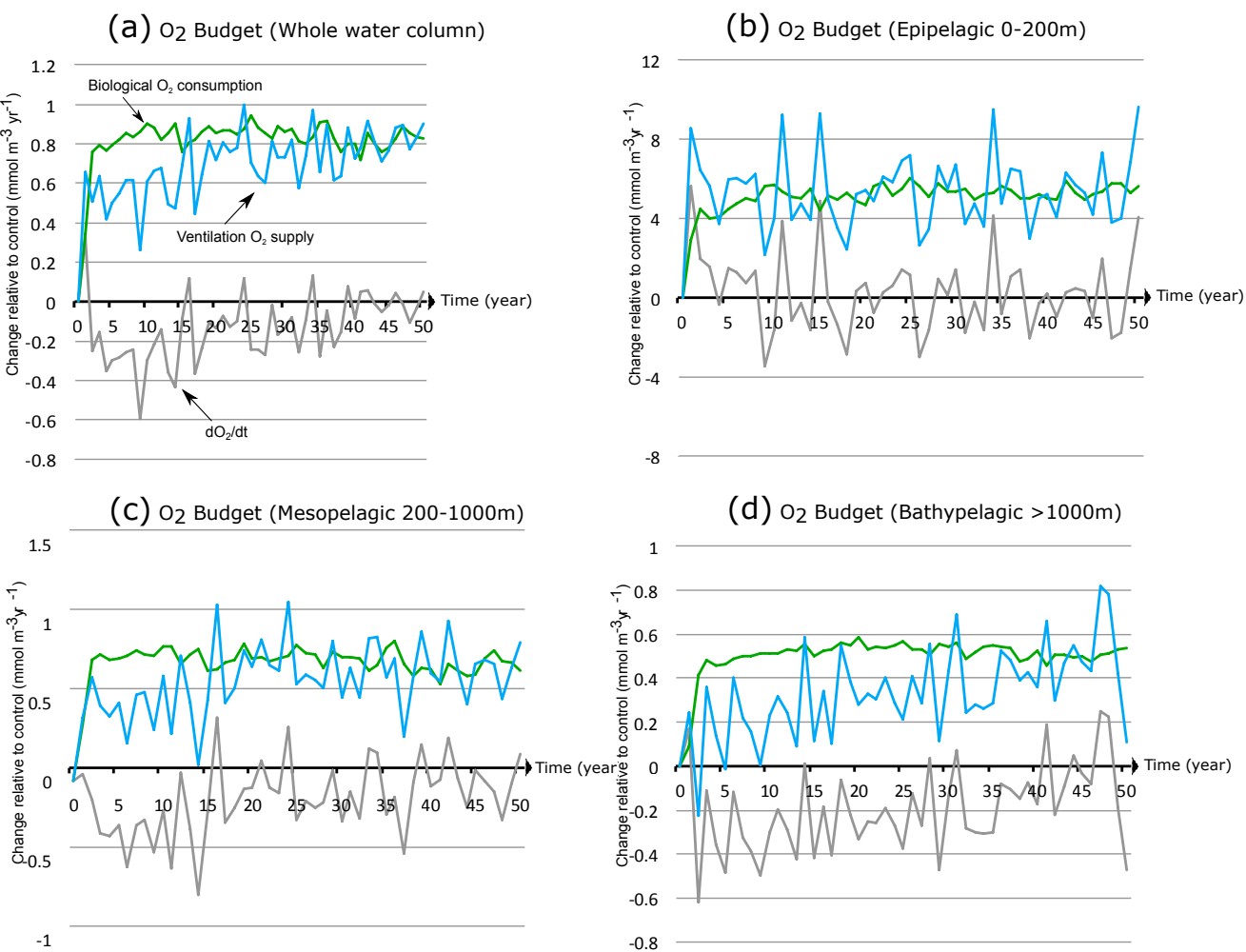

**Figure 9. Drivers of OMZ expansion under monsoon intensification.** Annual $O_2$ accumulation (tendency $\frac{dO_2}{dt}$) under increased wind stress (+50%) relative to control (grey line) and its biological (green) and physical (blue) sources in the OMZ as a function of time in (a) the whole water column, (b) the epipelagic zone (0-200m), (c) the mesopelagic zone (200-1000m) and (d) the bathypelagic zone (>1000m).

### 4.2.2 Implications for the nitrogen cycle

Denitrification increases by around 72% in response to a 50% increase in wind stress. This strong increase of denitrification amplifies fixed nitrogen removal. When integrated over the Arabian Sea domain, this represents an additional loss of fixed nitrogen of around 9.5 Tmol N/decade. Such a sink of bioavailable nitrogen can significantly alter the ocean productivity at large

5 scale. For instance, integrating the modern climate net primary production estimated from satellite data using the Behrenfeld and Falkowski (1997) VGPM across the Indian open ocean and assuming an average f-ratio of 0.5 (Singh and Ramesh, 2015; Prakash et al., 2015) yields an uptake of nitrate of around 25 Tmol N/year, with the Indian Monsoon Gyre and the Indian

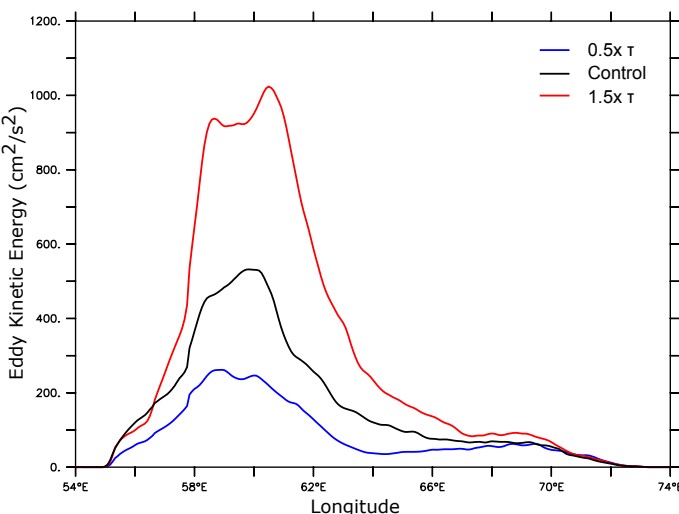

**Figure 10. Eddy Kinetic Energy as a function monsoon wind intensity.** Latitudinally averaged Eddy Kinetic Energy (in $cm^2\,s^{-2}$) between $17°N$ and $20°N$ as simulated under monsoon wind perturbations.

South Subtropical Gyres provinces (Longhurst, 2010) contributing by 13.28 and 11.78 Tmol N/year, respectively (see Table 1). Therefore, the enhanced denitrification in the Arabian Sea has the potential to significantly reduce biological productivity at the basin scale (and beyond) on timescales of decades to centuries. The extent of this reduction depends on the depth of the nitrogen removal and the local importance of other potential co-limitations. We speculate, however, that this perturbation would

eventually subside and weaken on timescales that approach the turnover time of fixed nitrogen in the ocean ($\sim$ 3000 years). This is because observations suggest a tight coupling between denitrification and $N_2$ fixation on timescales of thousand years (Gruber, 2008; Sigman and Haug, 2003). Two negative feedbacks may indeed limit, and eventually reverse, the growth of such a nitrogen cycle perturbation. First, the reduction in productivity that results from denitrification enhancement would ultimately reduce $O_2$ demand, and hence weaken the intensity of OMZ and denitrification. Second, the excess of phosphate over nitrate

(resulting from enhanced denitrification) would favor diazotrophic organisms that can outcompete normal phytoplankton in situations of severe fixed nitrogen deficits. This would lead to enhanced $N_2$ fixation and ultimately result in restoring the original balance (Gruber, 2008). Yet, there remain large uncertainties regarding the amplitude of these feedbacks and their timescales as other factors such as iron availability (Falkowski, 1997) can control $N_2$ fixation. Furthermore, observations of excess phosphate over nitrate indicate basin-scale imbalances between $N_2$ fixation and denitrification on timescales shorter

than the timescale of the overturning circulation.

The increase in the Arabian Sea denitrification and nitrification should also lead to an increase in the $N_2O$ production. This could not be tested in the present study as $N_2O$ is not represented in our model. Indeed, as denitrification leads to both production (under suboxic conditions) and consumption (under anoxic conditions) of $N_2O$, the net effect of a change in denitrification on $N_2O$ total budget is not easy to quantify without a dedicated parameterization of $N_2O$ fully taking into account

the different sources and sinks of the nitrous oxide as well as the effect of the transport and gas exchange on its dynamics.

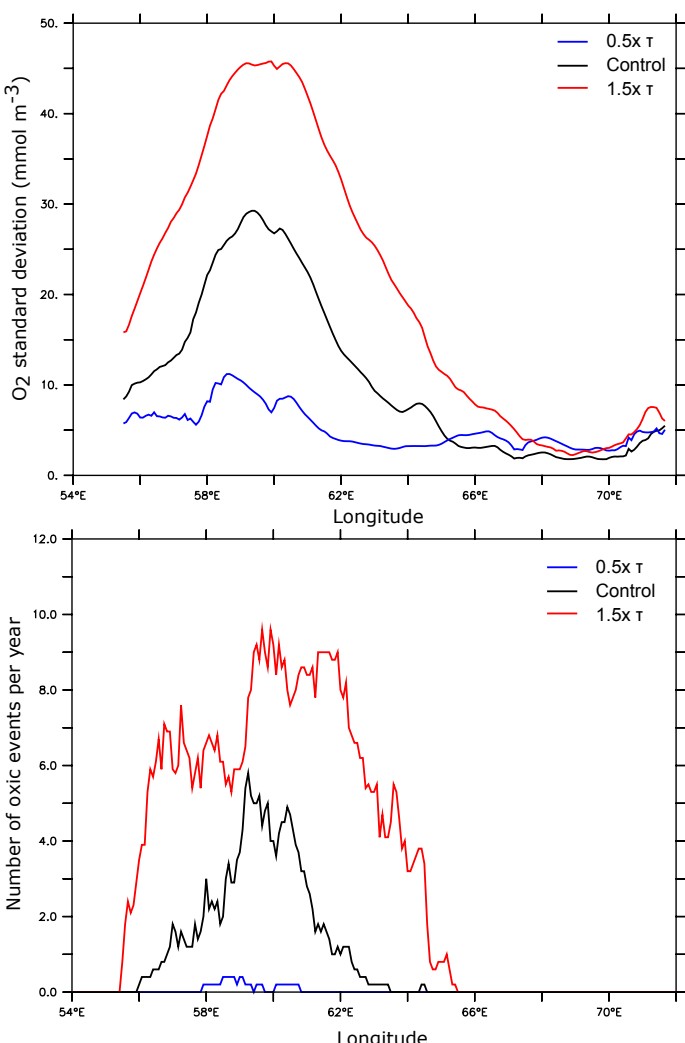

**Figure 11. Oxygen variability as a function of monsoon wind intensity.** (top) Oxygen temporal standard deviation (in mmol m$^{-3}$) and (bottom) number of episodic oxic events (O$_2$ > 60 mmol m$^{-3}$) per year along 17.5°N at 150 m (corresponding to the depth of the upper OMZ peak) as simulated under different monsoon wind intensities.

However, we speculate that significant monsoon intensification has the potential to lead to an important enhancement of N$_2$O production because of enhanced nitrification. Indeed, nitrification is predicted to increase by up to 62% in response to a 50% increase in wind stress while Arabian Sea data suggests nitrification to be the dominant pathway of N$_2$O production outside of the OMZ and a major contributor, together with denitrification, to its production inside the OMZ, (Bange et al., 2001).

5      In conclusion, reduced large-scale productivity under Indian monsoon (and denitrification) intensification could reduce the efficiency of the biological pump of carbon and together with enhanced N$_2$O production may amplify climate warming. There-

**Table 1.** Estimated biological productivity and nitrogen uptake in Longhurst (2010) Indian Ocean biophysical provinces. NPP is estimated from satellite data using the Behrenfeld and Falkowski (1997) Vertically Generalized Production Model (VGPM). We assume an average f-ratio (f-ratio = $\frac{NewP}{NPP}$ ) value of 0.5 consistent with previous studies of Singh and Ramesh (2015) and Prakash et al. (2015). NewP refers to new production.

| Longhurst (2009) biophysical province | Area (in km$^2$) | averaged NPP (in mg C m$^{-2}$ d$^{-1}$) | Integrated NPP (in Tmol N yr$^{-1}$) | Nitrate uptake (f-ratio=0.5) (in Tmol N yr$^{-1}$) |
|---|---|---|---|---|
| Indian Monsoon Gyres Province | 15797227 | 366.2 | 26.56 | 13.28 |
| Indian S. Subtropical Gyre Province | 16975241 | 324.8 | 23.56 | 11.78 |

fore, our study suggests that the Arabian Sea OMZ can contribute to climate variations over long timescales via a positive feedback on atmospheric concentrations of $CO_2$ and $N_2O$.

### 4.2.3 Implications for paleo studies

Our study reveals a strong link between the strength of the monsoon and the Arabian Sea productivity and denitrification. This validates the assumption made in several paleo-climate studies that rapid variations in productivity and denitrification in this region are tightly coupled to monsoon fluctuations. Our study also confirms the potential for the Arabian Sea OMZ to strongly impact the marine nitrogen budget at a larger scale in response to Indian monsoon fluctuations, in agreement with conclusions of previous paleo studies. Besides monsoon strength, some studies have linked past OMZ intensity changes to changes in the rate of formation and subduction of oxygen enriched Subantarctic Mode Water (SAMW) and Antarctic Intermediate Water (AAIW) in the Southern Ocean in association with Atlantic Meridional Overturning Circulation (AMOC) fluctuations (Pichevin et al., 2007; Böning and Bard, 2009). Previous model simulations of the last glaciation confirm that reduced AMOC leads to a reduction in nutrient supply to the Arabian Sea, thus resulting in a decreased productivity and a weakened OMZ (Schmittner et al., 2007). However, the simulated OMZ in the Arabian Sea is too weak in this very coarse resolution model, raising concern about its reliability in the region. In the Holocene, large-scale ventilation changes may have played an important role together with the fluctuations in monsoon intensity as suggested by recent studies (Rixen et al., 2014; Gaye et al., submitted; Das et al., 2017). For instance, recent paleo reconstructions by Das et al. (2017) suggest an intensification of the Arabian Sea OMZ from the middle to late Holocene despite a weakening of the SWM winds. These authors hypothesize that the recent ($\sim$ 4000 years BP onwards) decline in oxygen may have resulted from a cut off of the Arabian Sea OMZ from the oxygen enriched AAIW and SAMW. It is worth noting that these oxygen changes involve much slower timescales (ranging from several centuries to millennia) than the timescales associated with abrupt changes observed in paleo records (timescale of decades). Our study shows that the Arabian Sea OMZ can respond on a timescale of decades to changes in summer monsoon wind intensity, and hence strengthens the hypothesis that past abrupt changes in OMZ intensity are dominated by fluctuations in the intensity of monsoon winds. However, this hypothesis needs to be further confirmed through investigation using global simulations with more realistic representation of the Arabian Sea OMZ. Other studies proposed that weakened Arabian Sea OMZ during

stadials (cold phases) might be driven by both reduced productivity associated with weaker southwest (SW) monsoon as well as increased winter mixing associated with a stronger northeast (NE) monsoon (e.g. Reichart et al., 2004; Klöcker and Henrich, 2006). Here we show that the changes in the SW monsoon winds dominate the response of the Arabian Sea ecosystem and that the changes in the NE monsoon play a relatively smaller role. Therefore, our results validate previous paleo studies that

assign the dominant role of OMZ oscillations control to the Indian SW summer monsoon (e.g. Schulz et al., 1998; Altabet et al., 2002).

## 4.3   Sensitivity of the results to model resolution

Our model represents explicitly a large fraction of ocean eddies thanks to its relatively high-resolution of 1/12°. However, this resolution is still too coarse to resolve the full eddy spectrum. In particular, our model still overestimates the intensity of

the OMZ because it underestimates the ventilation of the northern Arabian Sea (Lachkar et al., 2016). As shown in Lachkar et al. (2016), eddies have a limited impact on the mean meridional circulation in the Arabian Sea. However, they do contribute significantly to the ventilation of the OMZ through enhanced eddy mixing along isopycnals. Here, we ask how resolution-dependent are the findings of the present study? To test the robustness of our results with respect to the employed model resolution, we examine the simulated response of NPP, denitrification and OMZ obtained using two different model resolutions:

1/12° and 1/3°. We find generally similar sensitivities to wind changes under both resolutions (Fig. 12). This is because the ventilation response to wind changes is identical in relative terms at both resolutions (Appendix A: Supplementary figures, Fig. A8). Indeed, at 1/12° resolution, the OMZ ventilation varies from around 3 sverdrup (Sv) when the wind stress is decreased by 50% to nearly 6 Sv when the wind stress is increased by 50%. At the substantially coarser resolution of 1/3°, this transport ranges from 2 Sv under decreased wind stress to nearly 4 Sv under increased wind stress (Appendix A: Supplementary figures,

Fig. A8). Therefore, the OMZ ventilation increases by nearly 50% as the resolution increases from 1/3° to 1/12°. However, the sensitivity of the ventilation to wind increase is very similar at both resolutions. We conclude that our findings are relatively robust with regard to the model resolution.

## 4.4   Caveats and limitations

### 4.4.1   Limitations of the model

The limitations of the model are among the study main limitations. For instance, our simulations are based on a simple biogeo-chemical model based on nitrogen only with no representation of other limiting nutrients such as iron, silicate and phosphate. We think this can lead to some local biases in regions where other nutrients can limit productivity (e.g., off the Somali coasts). However, at larger scales the limitation by nitrogen has been shown to dominate over other nutrient limitations in the Indian Ocean, and hence we assume that this choice should not affect the main findings and conclusions of the study (Koné et al.,

30   2009).

Another caveat of the study is the lack of a representation of the nitrogen fixation in the model. Although denitrification is thought to largely dominate over $N_2$ fixation in the Arabian Sea (Bange et al., 2005), the absence of nitrogen fixation in the

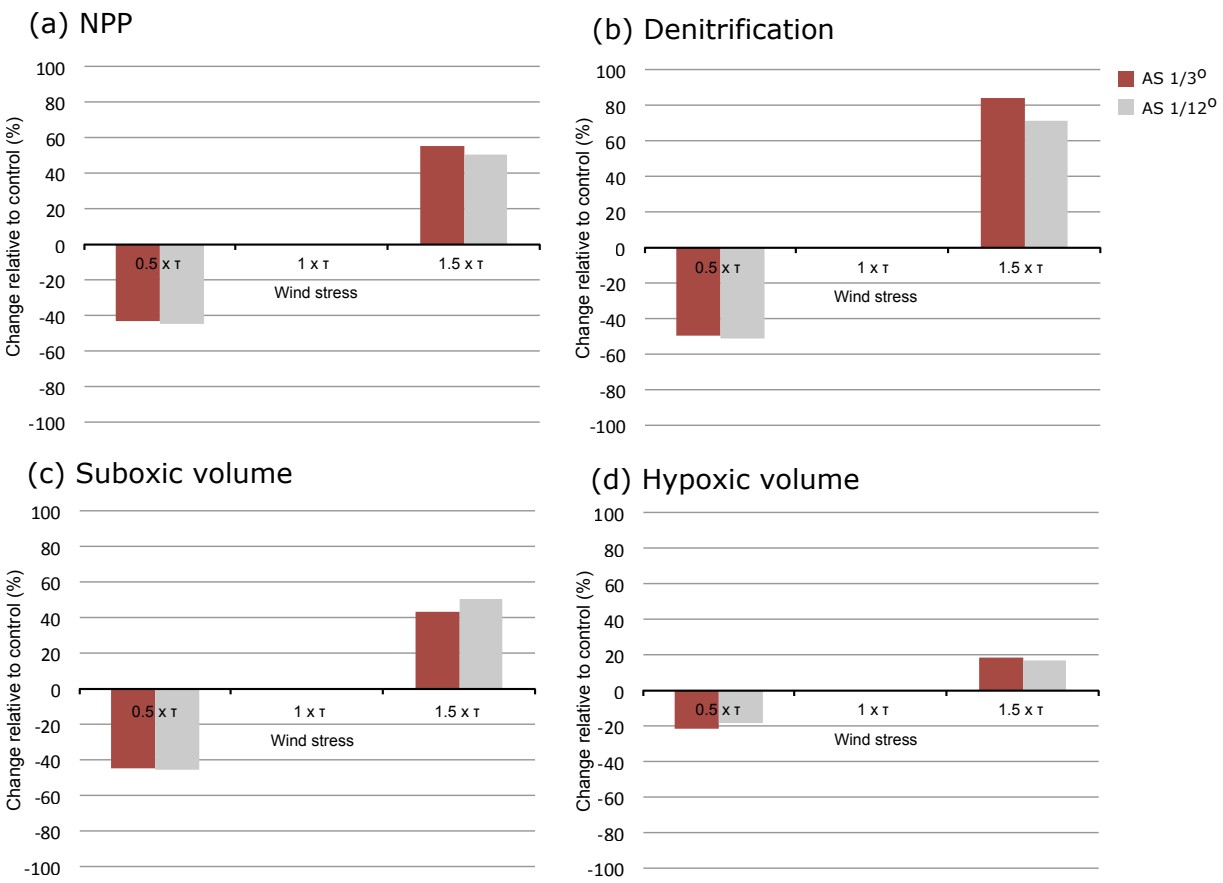

**Figure 12. Simulated response to monsoon wind changes as a function of model resolution.** Responses of (a) NPP, (b) denitrification, (c) suboxic volume and (d) hypoxic volume to monsoon wind changes as simulated using 1/3° (brown) and 1/12° (grey) model resolutions.

model could still artificially amplify the effect of denitrification on nitrogen budgets and ultimately lead to an overestimated feedback of OMZ fluctuations on the carbon cycle and the climate.

### 4.4.2 Limitations of the idealized wind change scenarios

The primary focus of this study is the sensitivity of the Arabian Sea OMZ to monsoon wind changes and its response timescale.

5   This justifies the use of highly idealized wind perturbations, as our simulations are not intended to mimic realistic past or future changes but rather to deepen our understanding of the key mechanisms at work and their potential implications. However, we are aware that future and past monsoon changes generally come in complex spatial and temporal patterns, which may affect the projected OMZ response. For instance, a recent study by Praveen et al. (2016) suggests that summer monsoon intensification will affect more the northern Arabian Sea (i.e., Sea of Oman) than its southern part (i.e., the Somali coast). This could alter

10   the response of the OMZ depending on how the overall Arabian Sea productivity changes. As an additional simplification, all

wind perturbations were applied instantly (following a step function). This is a strong idealization that can be justified here by the fact that we are interested in the equilibrium response of the ecosystem and its timescale. In reality, the response of the ecosystem will also depend on the perturbation timescale. To illustrate this, we performed two additional simulations where the wind stress was increased (resp. decreased) at a rate of 1%/yr over 50 years (Appendix A: Supplementary figures, Fig. A11). In these simulations, the changes in the size of the OMZ are consistent with runs where the perturbations are implemented instantly. However, the OMZ response is slower because of the slower perturbation timescale.

Finally, we considered here the effects of monsoon changes in isolation. The response of the OMZ to such perturbations may however change when considered in combination with other potential perturbations such as surface warming or large-scale ventilation changes. This needs to be further investigated in a dedicated study.

## 5  Conclusions

A set of coupled physical biogeochemical simulations of the Arabian Sea ecosystem reveals a tight coupling between the intensity of the summer monsoon wind and the size and intensity of the Arabian Sea OMZ. We find that the OMZ and ecosystem responses are largely determined by the perturbation of the summer SW monsoon, whereas the winter NE monsoon changes play a comparatively smaller role. We show that the intensification of monsoon winds strongly increases the ecosystem productivity, thereby amplifying the oxygen biological consumption and intensifying the OMZ at depth. Concurrently, increased monsoon winds also enhance the transport of oxygen to the OMZ. These opposing effects lead in the upper ocean to a weakening of the OMZ as the supply of oxygen through enhanced ventilation exceeds the oxygen depletion resulting from increased remineralization there. In contrast, the OMZ intensifies and expands at depth as the increased biological consumption of oxygen overcompensates the effect of enhanced ventilation below the thermocline. Our simulations indicate that the productivity responds to monsoon wind changes on a timescale of years, while the OMZ responds on a much longer timescale (i.e., several decades). This reflects the difference in the ocean circulation adjustment timescales between the surface and the intermediate ocean. The enhanced ventilation favors episodic injections of oxic waters in the lower epipelagic zone (100-200m) of the western and central Arabian Sea, leading to intermittent expansions of habitats and a more frequent alternation of hypoxic and oxic conditions there. The increased productivity and deepening of the OMZ also lead to a strong intensification of denitrification at depth, resulting in a substantial amplification of fixed nitrogen depletion in the Arabian Sea. We conclude that changes in the Indian monsoon can affect, on longer timescales, the large-scale biogeochemical cycles of nitrogen and carbon, with a positive feedback on climate change in the case of stronger winds. While it has been suggested that OMZs may expand in the future due to increased stratification (causing reduced ventilation), we show here that the Arabian Sea OMZ can also expand as a consequence of increased upwelling causing increased productivity and increased $O_2$ consumption due to enhanced remineralization. These results are obtained while considering the effects of monsoon wind changes in isolation. The response of the OMZ to wind increase may differ however in the presence of other concomitant perturbations such as potential changes in large-scale circulation and ventilation or additional surface warming.

**Appendix A: Supplementary figures**

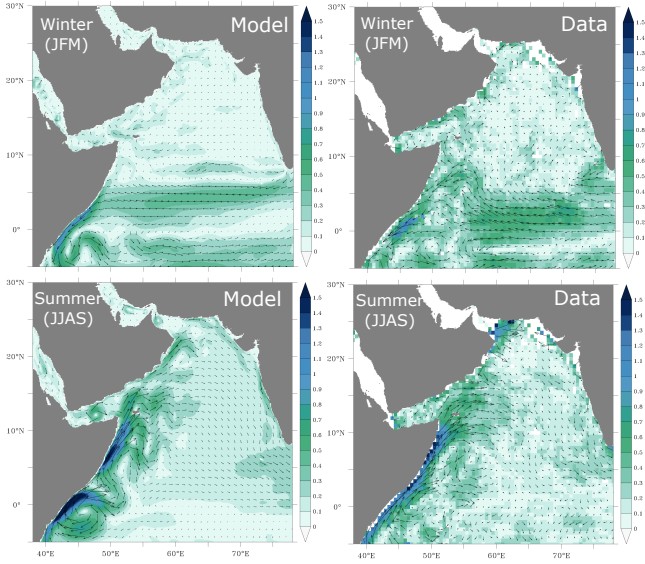

**Figure A1. Surface circulation.** Surface circulation as simulated in the model (left) and from surface drifter climatology of Lumpkin and Johnson (2013) (right) in winter (top) and summer (bottom) months. Arrows indicate the direction of the current and the color shading shows the current magnitude (in m/s).

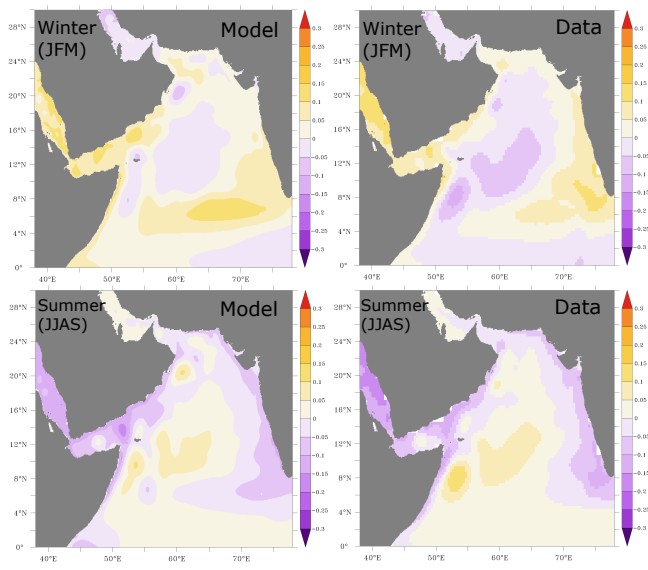

**Figure A2. Sea surface height anomalies.** Sea surface height seasonal anomalies (in m) simulated in ROMS (left) and from AVISO (right) during winter (top) and summer (bottom) months. The AVISO climatology is computed over the period from 1993 to 2009.

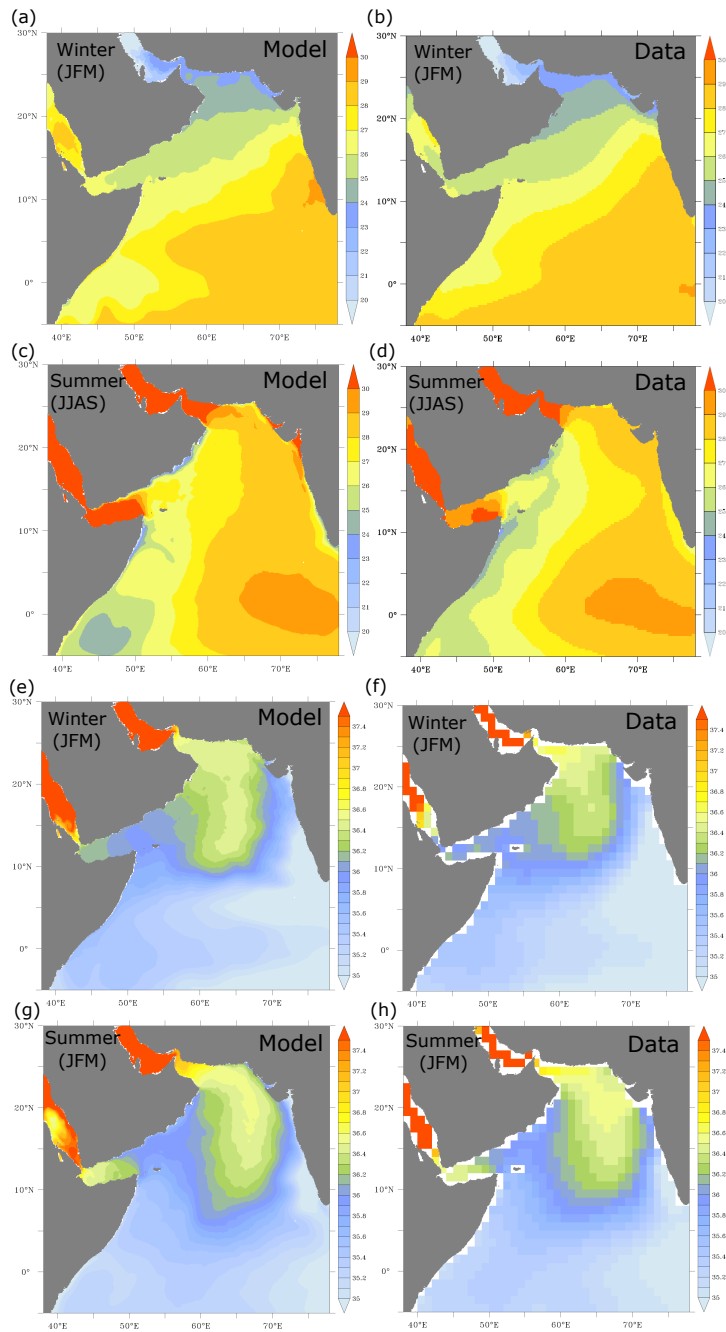

**Figure A3. Sea surface temperature and salinity.** (a-d) Sea surface temperature (in °C ) from ROMS (left) and AVHRR (right) during winter and summer months. (e-h) Sea surface salinity (in PSU) as simulated in ROMS (left) and from World Ocean Atlas (2009) dataset (right) during winter and summer months. The AVHRR observational climatology is computed over the period from 1981 to 2014.

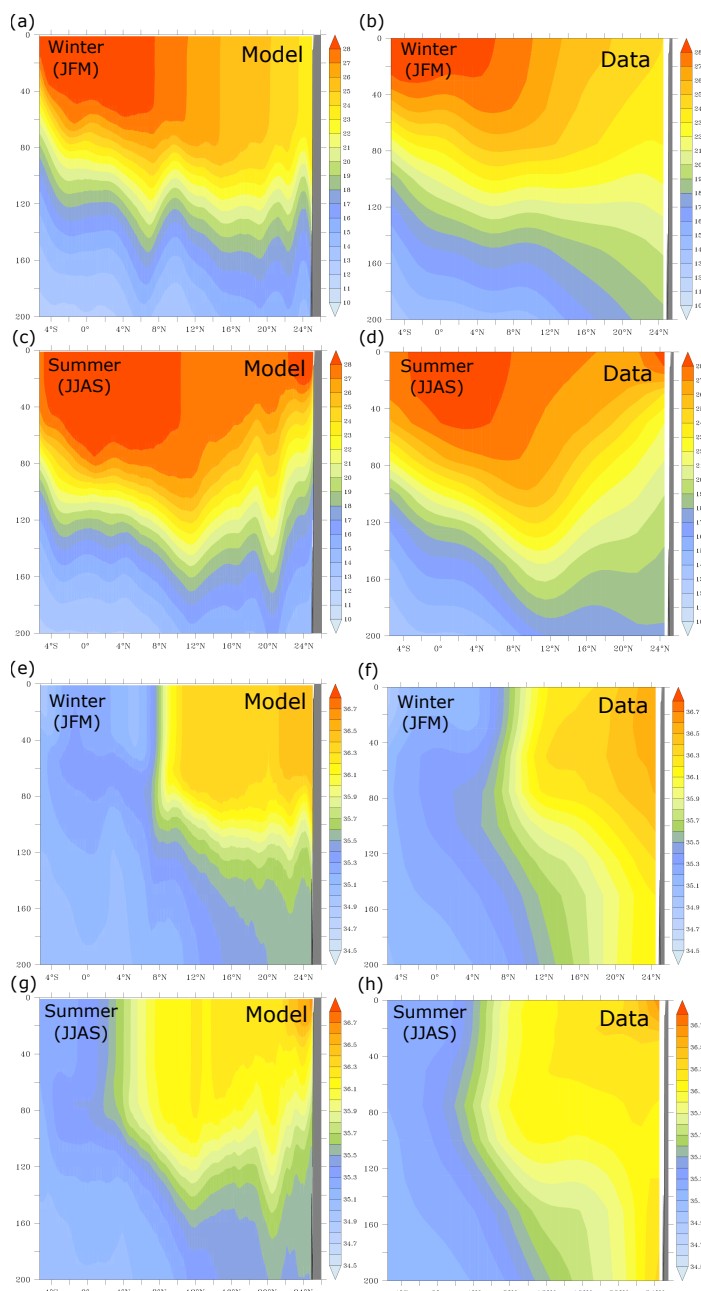

**Figure A4. Vertical distributions of temperature and salinity in the upper ocean.** (a-d) Meridional distribution of temperature in the upper 200 m along 62°E as simulated in ROMS (left) and from World Ocean Atlas (2009) dataset (right) during winter and summer months. (e-h) Meridional distribution of salinity in the upper 200 m along 62°E as simulated in ROMS (left) and from World Ocean Atlas (2009) dataset (right) during winter and summer months.

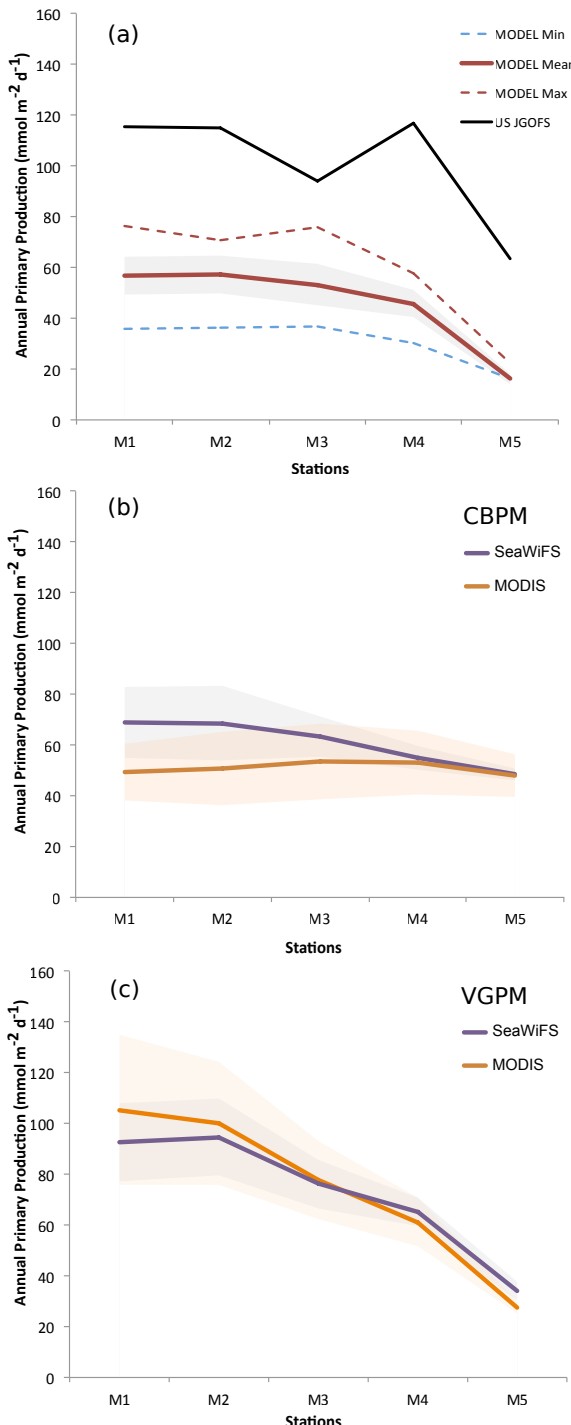

**Figure A5. Biological productivity at mooring stations M1 to M5 on a transect extending 1500 km from the coast of Oman.** (a) Annual-mean primary production fluxes at stations M1 to M5 (Lee et al, 1998) as estimated from in-situ observations (black) and simulated in the model (red). The dashed lines refer to modeled maximum (red) and minimum (blue) annual production. The shading indicates the ±1 standard deviation from the model mean. (b-c) Satellite-based estimates of annual-mean primary production fluxes at the M1 to M5 stations using the CBPM (b) and VGPM (c) algorithms and the SeaWiFS and MODIS sensors. The shading indicates the ±1 standard deviation from the mean.

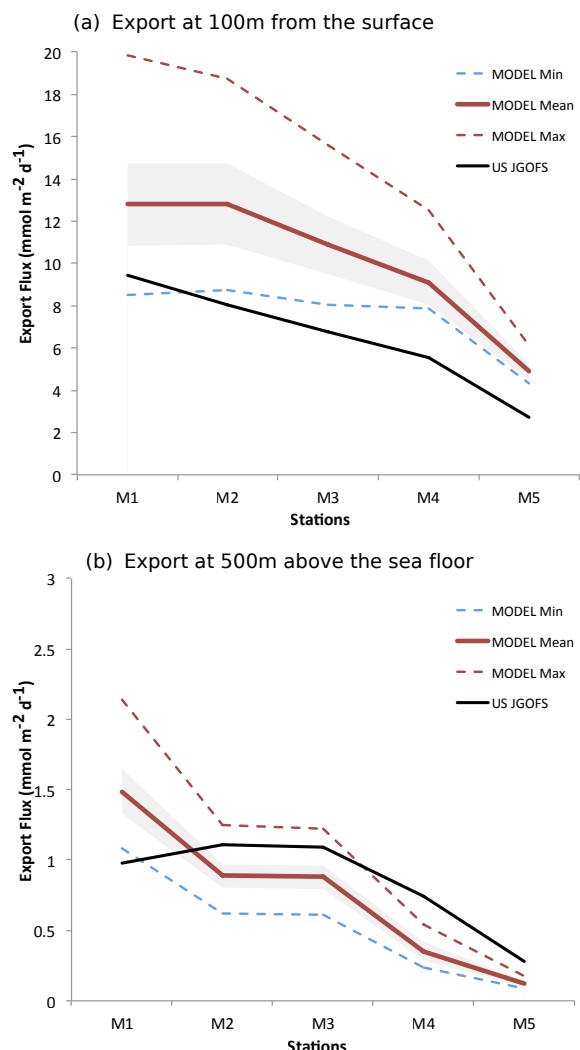

**Figure A6. Export fluxes at mooring stations M1 to M5.** Annual-mean export flux at the (M1-M5) stations as estimated from in-situ observations (black) and simulated in the model (red) at 100m (a) and 500m above the seafloor (b). The dashed lines refer to modeled maximum (red) and minimum (blue) annual export fluxes. The shading indicates the $\pm 1$ standard deviation from the model mean.

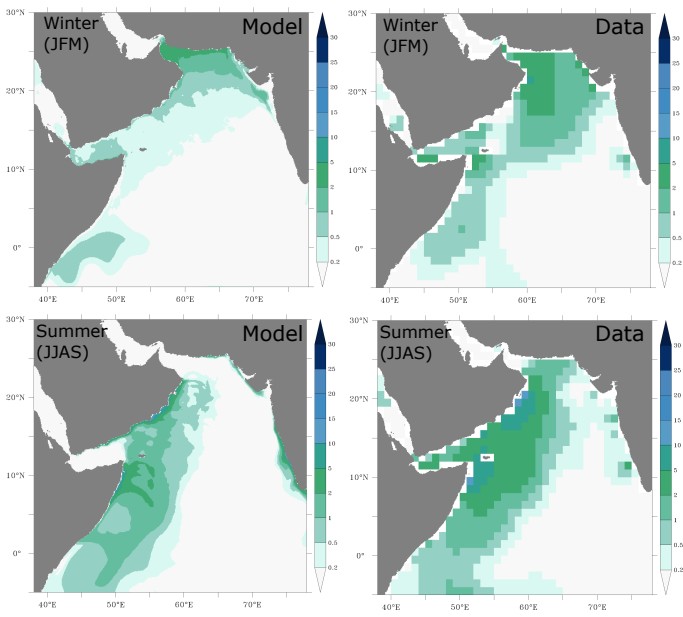

**Figure A7. Surface distribution of nitrate.** Surface distribution of nitrate (in mmol m$^{-3}$) as simulated in ROMS (left) and from World Ocean Atlas (2009) dataset (right) during winter (top) and summer (bottom) months.

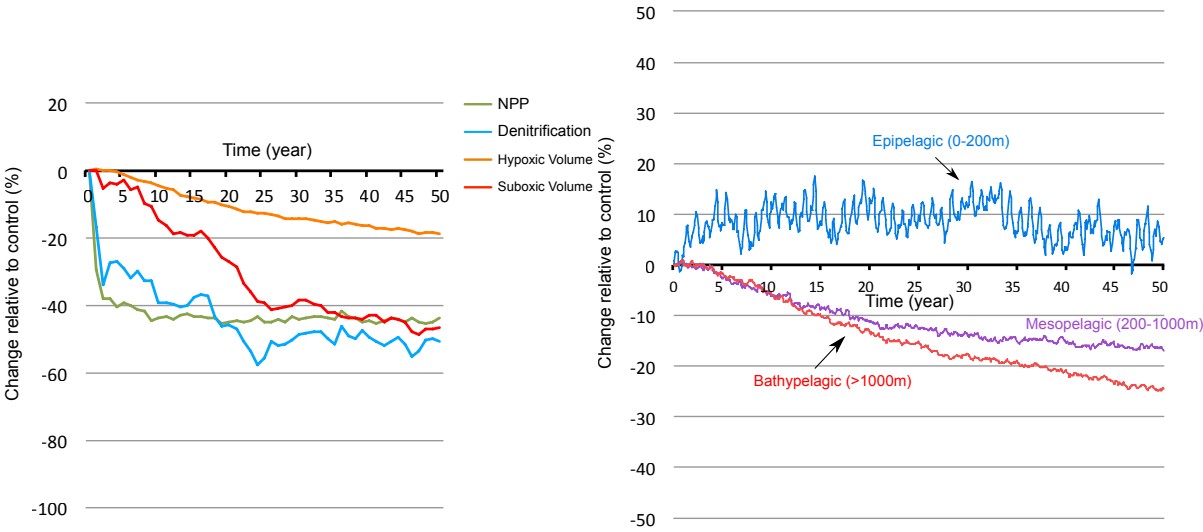

**Figure A8. Response to monsoon wind weakening.** (left) Response to wind stress decrease (-50%) of NPP, denitrification and suboxic and hypoxic volumes as a function of time. (right) Changes in hypoxic volume (in %) under wind stress decrease (-50%) in the epipelagic zone (0-200m), the mesopelagic zone (200-1000m) and the bathypelagic zone (>1000m).

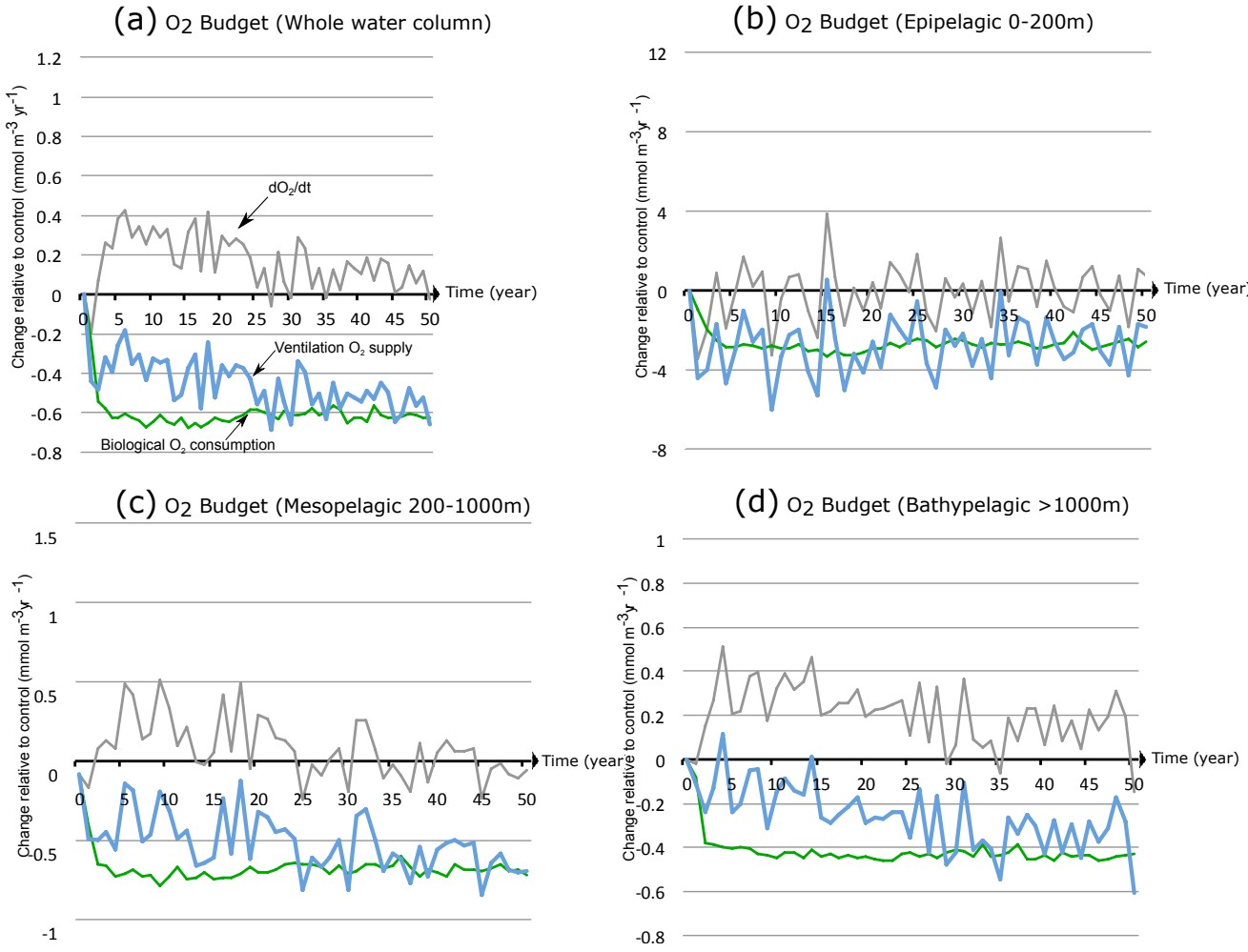

**Figure A9. Drivers of OMZ contraction under monsoon weakening.** Annual $O_2$ accumulation (tendency $\frac{dO_2}{dt}$) under decreased wind stress (-50%) relative to control (grey line) and its biological (green) and physical (blue) sources in the OMZ as a function of time in (a) the whole water column, (b) the epipelagic zone (0-200m), (c) the mesopelagic zone (200-1000m) and (d) the bathypelagic zone (>1000m).

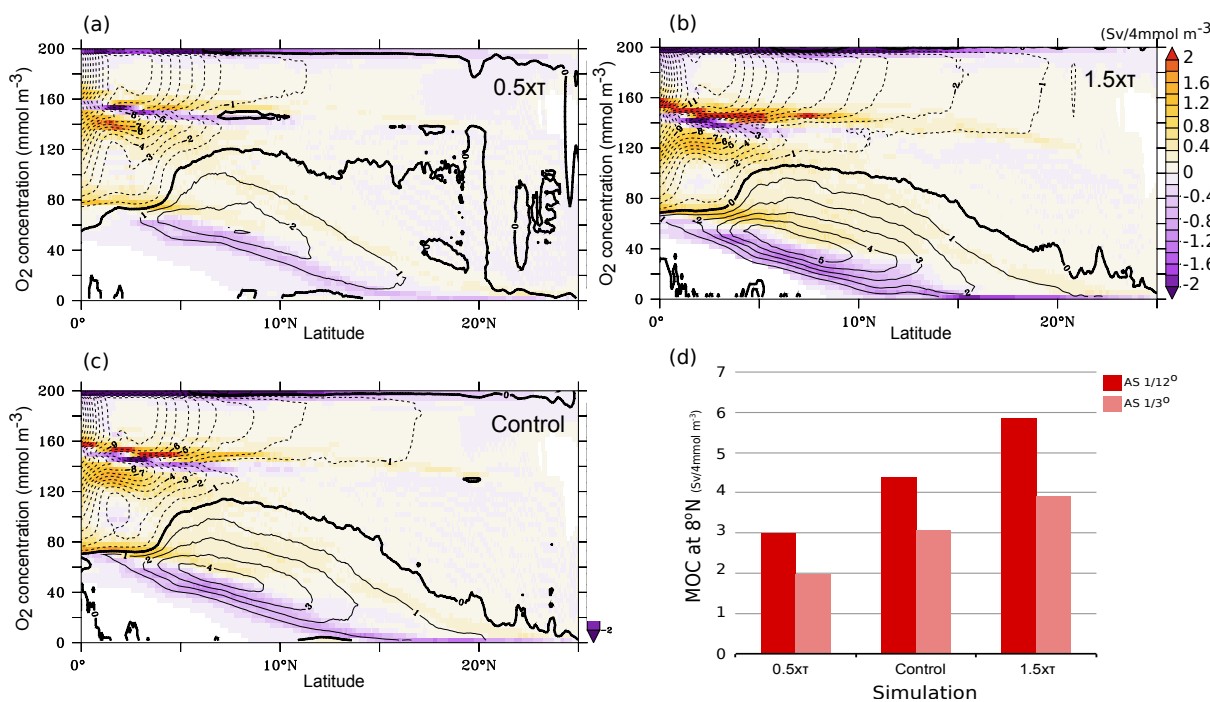

**Figure A10. Meridional transport in $O_2$ classes as a function of wind and resolution.** Meridional overturning circulation (MOC) in $O_2$ coordinate as a function of latitude (a-c) under different wind scenarios. Contour lines indicate the meridional stream function (positive indicates clockwise circulation). The color shading shows the meridional component of the transport (in Sv/ 4 mmol $O_2$ m$^{-3}$). (d) simulated MOC at 8°N under different wind scenarios using two model resolutions. We compute the meridional overturning circulation in oxygen coordinates following Lachkar et al. (2016). This approach allows us to characterize the ventilation of the OMZ driven by both the mean meridional circulation and eddy mixing along isopycnals. In practice this consists in binning the meridional transport by $O_2$ classes for each model resolution. The details of the method and its mathematical formulation are provided in Lachkar et al. (2016). Waters with intermediate oxygen concentrations that are above the hypoxic threshold enter the Arabian Sea at 8°N (latitude of the southern tip of India). This is balanced by a southward export of (i) highly oxygenated surface water ($O_2 > 160$ mmol m$^{-3}$) and (ii) hypoxic water ($O_2 < 60$ mmol m$^{-3}$) out of the Arabian Sea. The ventilation of the OMZ can be quantified in terms of the volume of hypoxic water exported out of the Arabian Sea.

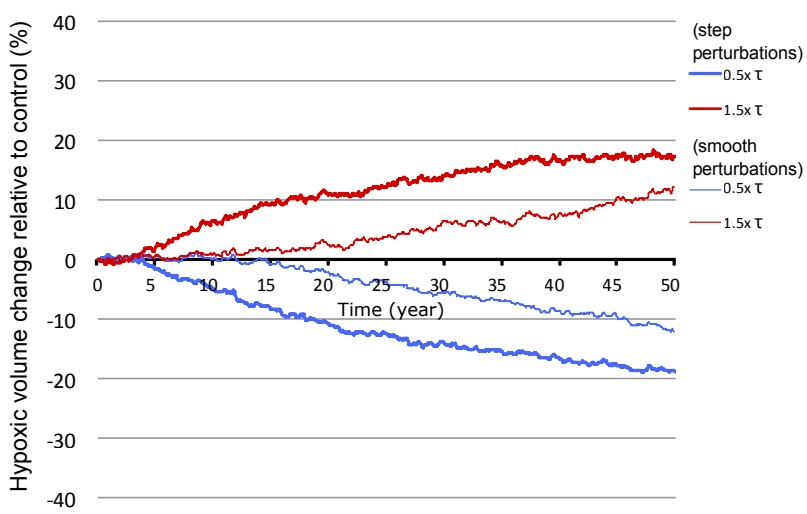

**Figure A11. OMZ response to gradual vs. abrupt wind changes.** Changes in hypoxic volume (in %) under gradually increasing or decreasing winds (thin lines) and instantly increased or decreased winds (thick lines). The smooth wind perturbation corresponds to an increase or decrease of wind stress at a rate of 1%/year over the 50-year duration of the simulation. The abrupt wind perturbation corresponds to an instant increase or decrease of wind stress by 50%.

## Appendix B: Code availability

The model code can be accessed online at: http://www.romsagrif.org. The model outputs are available from the authors upon request.

## Appendix C: Data availability

5  The data used for forcing and validating the model is publicly available online and can be accessed from cited references.

*Author contributions.* Z.L. conceived the study, performed the experiment and the analysis and wrote the manuscript. M.L. and S.S. contributed to the design of the study and participated in the interpretation of the results and the writing of the manuscript.

*Competing interests.* The authors declare that they have no competing financial interests.

*Acknowledgements.* Support for this research has come from the Center for Prototype Climate Modeling (CPCM), at New York University
10  Abu Dhabi (NYUAD). This research was carried out on the High Performance Computing (HPC) resources at NYUAD. We thank B. Marchand, M. Barwani and the whole NYUAD HPC team for technical support. We are thankful to N. Gruber for allowing access to the biogeochemical model code.

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
