# Peer review of "Intensification and deepening of the Arabian Sea Oxygen Minimum Zone in response to increase in Indian monsoon wind intensity"

_Biogeosciences, 2017_

## Referee Comment (RC1) · Anonymous Referee #1 · 1 Jun 2017

General Comments

This manuscript examines how changes in monsoon winds could impact the ocean ventilation, the biological activity and ultimately the oxygen minimum zone in the Arabian Sea. This work is based on an ocean regional model coupling ocean physics to biogeochemistry. This topic is crucial to our understanding of climate-induced changes in ocean biogeochemistry and the possible impacts for ecosystems and is highly relevant for Biogeosciences. The future of the Arabian Sea's OMZ is still unclear. Available observations of the past decades are too sparse to get a full picture in this region and previous modeling studies either did not capture the main features of this OMZ (coarse

resolution climate models) or did not cover long enough periods to tackle this issue. This study, although idealized in the monsoon wind changes, gives perspective on the changes to be expected in the Arabian Sea.

I really enjoyed reading this manuscript. The approach is sound, the results are clearly presented (figures and text), the authors analyzed extensively the processes at play using numerous sensitivity model experiments and discussed the implications and limitations of their results.

I recommend this manuscript for publication in Biogeosciences. Nevertheless, I have a few comments, mostly about the discussion. In particular, I would like to see the results on the denitrification placed in a broader and global context (comment #1). I also would like to see a slight increment in the discussion on the relative role of NEM vs. SWM (comment #3). Finally, I have a question about the discussion of N2O (comment #2).

Specific Comments

1) P15, P17 and other places in the manuscript: "On the other hand, the changes in the OMZ intensity have the potential - via denitrification - to alter the marine nitrogen budget, and hence the efficiency of the biological pump of carbon and climate, on the longer timescales." "Therefore, the enhanced denitrification in the Arabian Sea has the potential to significantly reduce biological productivity at the basin scale (and beyond) on timescales of decades to centuries."

We usually consider that on long time scales, denitrification and nitrogen fixation compensate each other at the global scale. Water masses where denitrification occurs at depth present an excess in available phosphate. When this excess in phosphate makes it back to the surface it can support nitrogen fixation. Could you please discuss your result in this context? On what temporal and spatial scales is your result pertinent? Do you expect a global compensation of this increase in denitrification on longer timescales? How would this impact your conclusion on biological productivity, locally and globally? You briefly discuss the limitation of not having nitrogen fixation in

your model but my comment here is more general and calls for some discussion and perspectives on how your results fit in the more global climate change context.

2) In P18, you discuss the production of N2O. Based on previous work on O2 and N2O production, could you compute a first order back of the envelope estimate of how much N2O could be produced by your O2 changes? How does that compare to previous estimates and to the global production of N2O in the ocean and out of the ocean?

3) P19: "Here we show that the changes in the SW monsoon winds dominate the response of the Arabian Sea ecosystem and that the changes in the NE monsoon play a relatively smaller role. Therefore, our results validate previous paleo studies that assign the dominant role of OMZ oscillations control to the Indian SW summer monsoon (e.g. Schulz et al., 1998; Altabet et al., 2002)."

You should discuss why the dominance of the SWM is to be expected: 1) the biological production during the SWM dominates the total annual production and 2) in your model NEM winds primarily increase MLD, ventilation and provides O2 to the region, as shown by the higher increase in the suboxic volume in your SWM+/NEM- simulation than in your SWM+/NEM+ simulation (Fig 5).

Technical Corrections

Figure 4: could you make the numbers on panel b more visible.

---

## Referee Comment (RC2) · Anonymous Referee #2 · 13 Jun 2017

The authors used a ROMS, which was coupled to an NPZD model to study impacts of changing monsoon winds on the OMZ and the marine nitrogen cycle in the Arabian Sea. The results indicate that changes in the summer monsoon winds exert the main control on productivity, the OMZ and finally the marine nitrogen cycle. Intensification of the summer monsoon winds increases the productivity, expands the OMZ at depth, and increases denitrification, while an enhanced intrusion of oxygen-enriched surface water weakens the intensity of the upper OMZ at water-depth between 100 and 200 m. Since there are indications that the Indian summer monsoon intensifies in response to global warming, the topic addressed within the manuscript is of great relevance. The manuscript is, moreover, well-written. However, the presented model results and

parameterizations of important processes deviate from conclusions drawn from field data. This in addition to some other aspects needs clarification before publication of the manuscript can be recommended.

As stated in the abstract the main conclusion is as follows: 'We show that the Arabian Sea productivity increases and its OMZ expands and deepens in response to monsoon wind intensification. These responses are dominated by the perturbation of the summer monsoon wind, whereas the changes in the winter monsoon wind play a secondary role'. Here it should be mentioned explicitly that winds are generally weak and winter cooling drives productivity during the winter monsoon (e.g. Madhupratap et al. 1996). In its present form it is misleading because it could imply that wind mixing is a dominant factor because it was selected to run the sensitivity experiment. This assumption would furthermore suggest that model results show that the summer monsoon is more important for the productivity as the winter monsoon. The discussion of various paleooceanographic studies shows that warming increases wind speeds, expands the OMZ and increases denitrification. This, furthermore supports the impression that the summer monsoon is the main driver, and the winter monsoon of lower importance. This was not studied in the model and it should also be considered that these paleoceanographic results were obtained by comparing glacial and interglacial periods. During the Holocene a weakening of the summer monsoon strength seems to be accompanied by an intensification the OMZ (see e.g. Rixen et al. 2014) suggesting that ventilation plays a more important role than implied by the model output.

2) The occurrence of the secondary nitrite maximum is generally assumed to indicate active denitrification in the water column of the Arabian Sea (see Naqvi et al. 1991, 1998 and more recently Bulow et al., 2010, Gaye et al. 2013). The secondary nitrite maximum occurs a water depth between 100 and 400 m which implies that denitrification is absence or at least of minor importance in the deeper part of the OMZ. The model results show exactly the opposite as summarized in the abstract: 'The increased productivity and deepening of the OMZ also lead to a strong intensification of denitrification at depth, resulting in a substantial amplification of fixed nitrogen depletion in the Arabian Sea'. This needs to be clarified as well as the ignored N-fixation as pointed out by reviewer #1.

3) The parameterization of the carbon export into the deep sea should be described in more detail. Since sinking speeds and respiration rates are provided I assume that a model similar to those introduced by Banse (1990) was used. The considered sinking speeds of 1 and 10 m per day are an order of magnitude lower as those derived from sediment trap studies (see e.g. Berelson, 2001). Please clarify.

4) Among others satellite-derived chlorophyll concentrations were used to validate the model, which to my understanding do not agree well to model outputs. (The months given in Fig. 2 bottom need to be corrected). However, satellite data especially during the summer monsoon are problematic but there are a number of sediment trap data from the Arabian Sea (see e.g. Honjo et al. 1997 and Lee et al. 1997). Considering the importance of carbon export model data should be compared to sediment trap data to make the main conclusions convincing.

5) Considering the overall importance of the selected topic, which will probably attract a wider readership, I recommend to avoid Taylor diagrams and use simple xy scatter plots. They are clear and easy to interpret. Please include also data from the deeper part of the OMZ in the data / model comparison.

6) Moel et al. 2009 is missing in the reference list

---

## Author Comment (AC2) · 25 Jul 2017

**Answer to Referee #2**

The authors would like to thank anonymous referee #2 for the valuable comments and suggestions, which will certainly help to improve the manuscript. A detailed point-by-point reply to the comments follows below, where reviewer comments are slanted and author responses are blue.

**Anonymous Referee #2**

*The authors used a ROMS, which was coupled to an NPZD model to study impacts of changing monsoon winds on the OMZ and the marine nitrogen cycle in the Arabian Sea. The results indicate that changes in the summer monsoon winds exert the main control on productivity, the OMZ and finally the marine nitrogen cycle. Intensification of the summer monsoon winds increases the productivity, expands the OMZ at depth, and increases denitrification, while an enhanced intrusion of oxygen-enriched surface water weakens the intensity of the upper OMZ at water-depth between 100 and 200 m. Since there are indications that the Indian summer monsoon intensifies in response to global warming, the topic addressed within the manuscript is of great relevance. The manuscript is, moreover, well-written. However, the presented model results and parameterizations of important processes deviate from conclusions drawn from field data. This in addition to some other aspects needs clarification before publication of the manuscript can be recommended.*

We are thankful to the reviewer #2 for the time spent on reviewing our manuscript and for his/her valuable comments that will make our manuscript stronger. Following the reviewer suggestion, we will add model comparisons with field data to the revised the manuscript to further support our main results. Moreover, we will add a couple of clarifications as requested by the reviewer. Please see below our responses to specific comments.

*1) As stated in the abstract the main conclusion is as follows: 'We show that the Arabian Sea productivity increases and its OMZ expands and deepens in response to monsoon wind intensification. These responses are dominated by the perturbation of the summer monsoon wind, whereas the changes in the winter monsoon wind play a secondary role'. Here it should be mentioned explicitly that winds are generally weak and winter cooling drives productivity during the winter monsoon (e.g. Madhupratap et al. 1996). In its present form it is misleading because it could imply that wind mixing is a dominant factor because it was selected to run the sensitivity experiment.*

We agree with the reviewer that winter winds are generally weak and that winter cooling and convection drives winter bloom. We will make this even more explicit in the revised manuscript. We will also explain better the mechanisms through which the perturbation of the summer monsoon controls the OMZ annual mean response (please also see our response to comment #3 by reviewer#1).

*This assumption would furthermore suggest that model results show that the summer monsoon is more important for the productivity as the winter monsoon. The discussion of various pale-oceanographic studies shows that warming increases wind speeds, expands the OMZ and increases denitrification. This, furthermore supports the impression that the summer monsoon is the main driver, and the winter monsoon of lower importance. This was not studied in the model and it should also be considered that these paleoceanographic results were obtained by comparing glacial and interglacial periods. During the Holocene a weakening of the summer monsoon strength seems to be accompanied by an intensification the OMZ (see e.g. Rixen et al. 2014) suggesting that ventilation plays a more important role than implied by the model output*

In response to this comment, we will improve our discussion of the mechanisms that lead to stronger control by the summer monsoon perturbation (please see our response to previous comment). We would also like to point out that our finding that the summer monsoon driven productivity exceeds that of the winter monsoon is also supported by several observations (e.g., Dickson et al, 2001). Otherwise, we agree that past ventilation changes may have played an important role in modulating the variations in the Arabian Sea OMZ and denitrification as suggested by some previous studies (Pichevin et al, 2007, Boning & Bard, 2009) and already acknowledged in our manuscript (see section 4.2.3 of the manuscript). While our study highlights the strong link between monsoon variations and OMZ fluctuations, it does not rule out a potential contribution from changes in large-scale ventilation. With our current model setup we cannot however test such a hypothesis as this would require using global simulations with a realistic representation of the Arabian Sea OMZ as stated in the submitted manuscript (see section 4.2.3, page 18).

*2) The occurrence of the secondary nitrite maximum is generally assumed to indicate active denitrification in the water column of the Arabian Sea (see Naqvi et al. 1991, 1998 and more recently Bulow et al., 2010, Gaye et al. 2013). The secondary nitrite maximum occurs a water depth between 100 and 400 m which implies that denitrification is absence or at least of minor importance in the deeper part of the OMZ. The model results show exactly the opposite as summarized in the abstract: 'The increased productivity and deepening of the OMZ also lead to a strong intensification of denitrification at depth, resulting in a substantial amplification of fixed nitrogen depletion in the Arabian Sea'. This needs to be clarified as well as the ignored N-fixation as pointed out by reviewer #1.*

We do not agree with the reviewer statement in that our modeled denitrification profile disagrees with observations. Indeed, our control simulation also shows maximum denitrification between 100 and 400m (black curve in Fig. 7b of the submitted manuscript). For example the rate of simulated denitrification in the control run below 400m is at least a factor 7 smaller than at 200m. Our results are therefore consistent

with observations made by studies cited by the reviewer (e.g., Bulow et al., 2010, Gaye et al. 2013). The deepening of denitrification referred to in the statement cited by the reviewer concerns the 50% increased wind perturbation simulation. We do not expect the model subjected to such a relatively extreme perturbation to stay close to observations made under present day forcing.

*3) The parameterization of the carbon export into the deep sea should be described in more detail. Since sinking speeds and respiration rates are provided I assume that a model similar to those introduced by Banse (1990) was used. The considered sinking speeds of 1 and 10 m per day are an order of magnitude lower as those derived from sediment trap studies (see e.g. Berelson, 2001). Please clarify.*

The detail of the representation of the carbon export in the model is given in section 2.1, lines 18-22, page 4. Following the lead of Gruber et al (2006), particle sinking is represented explicitly using 2 detritus classes that can also be advected laterally: a class of large and fast sinking particles (10m d$^{-1}$) and another class of small and slow sinking particles (1m d$^{-1}$). We also specify the remineralization rates used for the large (0.01 d$^{-1}$) and small detritus (0.03 d$^{-1}$). We do not use representations of export based on the Martin equation where the particle flux is set to decrease exponentially with depth such the ones referred to by the reviewer (and described in Banse 1990 or Berelson, 2001). Instead, the flux attenuation with depth emerges from the decomposition of organic matter as it sinks.

We would like to point out that it is the ratio of sinking speed to decomposition rate (corresponding to a remineralization lengthscale) that controls the attenuation of export fluxes in our model. While sinking speeds used in the model can be lower than some sediment trap estimates by up to one order of magnitude as correctly mentioned by the reviewer, the decomposition rates used in the model are also proportionally weaker than in these studies (e.g., ~ 0.2 to 0.3 d$^{-1}$ in Banse 1990, Deep Sea Research). Therefore, despite differences in sinking speed and decomposition rates, the remineralization lengthscales in our model (1000m and 33m for large and small detritus, respectively) are comparable to those implied in some previous studies. This is supported by the reasonable agreement of our simulated export fluxes with the sediment trap observations from the US JGOFS Arabian Sea expedition (see the new Figure 3 in our response to comment #4 below).

*4) Among others satellite-derived chlorophyll concentrations were used to validate the model, which to my understanding do not agree well to model outputs. (The months given in Fig. 2 bottom need to be corrected). However, satellite data especially during the summer monsoon are problematic but there are a number of sediment trap data from the Arabian Sea (see e.g. Honjo et al. 1997 and Lee et al. 1997). Considering the importance of carbon export model data should be compared to sediment trap data to make the main conclusions convincing.*

We agree with the reviewer that the modeled chlorophyll-a does not agree well with the observations in certain areas, especially off the coast of Somalia as already acknowledged in the submitted manuscript (lines 9-10, page 8). However, the fidelity of the model north of 10ºN is in line (if not better) with most of state of the art models (e.g., Resplandy et al, 2012). This is also supported by the relatively high correlations and comparable variances between simulated and observed surface chlorophyll-a distributions evidenced in the Taylor diagrams (Fig 4).

Yet, we do agree with reviewer #2 comment that satellite chlorophyll data is not enough to evaluate the biological model and we thank him/her for his/her suggestion to include more field data in the model validation. Following the reviewer suggestion, we invested time to enhance our model evaluation by adding comparisons with field data from the US JGOFS Arabian Sea Process Study (1995). This consists in: i) $^{14}$C primary productivity ii) export fluxes at 100m estimated using $^{234}$Th removal rates and iii) export fluxes estimated from sediment trap data at 500m above the seafloor to avoid including resuspension fluxes as advised in previous works (e.g., Gardener 1992). Fluxes were measured essentially during the year 1995 at 5 sites (M1, M2, M3, M4 and M5) along a transect extending from the Coast of Oman to the central Arabian Sea (a figure from Lee et al. (1998) indicating the location of the sediment trap moorings is included at the end of this document). Because of the relatively limited number of individual in-situ observations of biological productivity available in this dataset (only 5 measurements at each site), we also used satellite-based productivity estimates obtained using two different algorithms: the Vertically Generalized Production Model (VGPM) (Behrenfeld and Falkowski, 1997a) and the Carbon Based Production Model (CBPM) (Westberry et al., 2008) using data from two sensors (SeaWiFS and MODIS). The results of these comparisons are presented in the following figures.

[Figure]

Figure 1: Annual-mean primary production fluxes at 5 mooring stations (M1-M5) as estimated from in-situ observations (black) and simulated in the model (red). The dashed lines refer to modeled maximum (red) and minimum (blue) annual production. The shading indicates the ± 1 standard deviation from the model mean.

[Figure]

Figure 2: Satellite-based estimates of annual-mean primary production fluxes at the 5 mooring stations (M1-M5) using the CBPM (top) and VGPM (bottom) algorithms and the SeaWiFS and MODIS sensors. The shading indicates the ± 1 standard deviation from the mean.

[Figure]

[Figure]

Figure 3: Annual-mean export flux at 5 mooring stations (M1-M5) as estimated from in-situ observations (black) and simulated in the model (red) at 100m (top) and 500m above the seafloor (bottom). The dashed lines refer to modeled maximum (red) and minimum (blue) annual export fluxes. The shading indicates the ± 1 standard deviation from the model mean.

This comparison shows that the model correctly simulates a decrease in productivity and export fluxes as the distance to the coast increases (Figures 1 and 3). The model, however, substantially underestimates the measured primary productivity in all 5 stations. Some of this mismatch may be due to the fact that the in-situ productivity estimates are all coming from one individual year (1995) and based on only 5

independent measurements at each site (Lee et al, 1998). Given the importance of both mesoscale and interannual variability, the in-situ estimates may therefore not be representative of the long-term climatological conditions simulated by the model. Indeed, by further comparing the simulated productivity to satellite-based estimates at the 5 stations, we find generally a better agreement (Figure 2). We further contrasted the simulated export fluxes at 100m to estimates from Lee et al (1998) at the 5 stations (Figure 3). Our modeled export fluxes generally overestimate the [234]Th-based estimates but remain comparable in magnitude with these observations. Furthermore, the model reproduces quite accurately the observed offshore gradient in export. It is worth highlighting however that similarly to in-situ measured productivity, these export fluxes are based on 4 independent measurements at each site only, all from the same year. This may induce biases in these estimates due to contamination by mesoscale and interannual variability. We finally compared the modeled export fluxes in the deep ocean (500m above the seafloor) to sediment trap data at the same 5 sites (Figure 3). The comparison shows a good agreement between the model and the observations at all stations. It is worth noting that these deep export flux estimates can be considered as more robust than those at 100m as they are based on a larger number of independent measurements (20-40 measurements at each site).

In conclusion, despite some discrepancies, our modeled fluxes show a reasonable agreement with both field data and satellite observations. Following reviewer's suggestion, we will include the 3 figures presented above together with a description of these comparisons in the revised manuscript. We will also correct the typo in the months given in Fig 2 of the submitted manuscript.

*5) Considering the overall importance of the selected topic, which will probably attract a wider readership, I recommend to avoid Taylor diagrams and use simple xy scatter plots. They are clear and easy to interpret. Please include also data from the deeper part of the OMZ in the data / model comparison.*

We prefer the Taylor diagrams over xy plots because the former provide a more quantitative and condensed synthesis of model skill. As each dot on the Taylor diagrams represents an independent comparison between the model and the data, replacing the two diagrams with xy plots would require 18 figures! Additionally, because of the large number of individual observations used in this comparison (resulting from the high-resolution of the satellite products and the large number of observations available in World Ocean Database), the xy plots may be visually difficult to read and compare. Therefore, we decided to keep the Taylor diagrams as they are widely used for model evaluation and model skill assessments in climate and environmental sciences. Please also note that Fig 4b (submitted manuscript) include observations sampled down to 1000m (grey filled circles).

*6) Moel et al. 2009 is missing in the reference list.*

Thank you, we will correct this.

[Figure]

Fig. 1. Map showing location of sediment-trap moorings 1–5.

Figure 1 from Lee et al, (1998, Deep Sea Research II) indicating the location of sediment traps M1 to M5.

---

## Author Response (AR1)

The authors would like to thank both anonymous referees for their valuable comments and suggestions, which have helped improving the manuscript. A detailed point-by-point reply to the comments follows below, where reviewer comments are slanted and author responses are blue.

**Anonymous Referee #1**

*General Comments*

*This manuscript examines how changes in monsoon winds could impact the ocean ventilation, the biological activity and ultimately the oxygen minimum zone in the Arabian Sea. This work is based on an ocean regional model coupling ocean physics to biogeochemistry. This topic is crucial to our understanding of climate-induced changes in ocean biogeochemistry and the possible impacts for ecosystems and is highly relevant for Biogeosciences. The future of the Arabian Sea's OMZ is still unclear. Available observations of the past decades are too sparse to get a full picture in this region and previous modeling studies either did not capture the main features of this OMZ (coarse resolution climate models) or did not cover long enough periods to tackle this issue. This study, although idealized in the monsoon wind changes, gives perspective on the changes to be expected in the Arabian Sea.*

*I really enjoyed reading this manuscript. The approach is sound, the results are clearly presented (figures and text), the authors analyzed extensively the processes at play using numerous sensitivity model experiments and discussed the implications and lim- itations of their results.*

*I recommend this manuscript for publication in Biogeosciences. Nevertheless, I have a few comments, mostly about the discussion. In particular, I would like to see the results on the denitrification placed in a broader and global context (comment #1). I also would like to see a slight increment in the discussion on the relative role of NEM vs. SWM (comment #3). Finally, I have a question about the discussion of N2O (comment #2).*

We are grateful to the reviewer #1 for the time spent on reviewing our manuscript and for his/her positive and insightful comments that have helped improving the quality of the manuscript. We have revised the manuscript to improve the discussion of the three points raised by the reviewer. Please see below our responses to specific comments.

*Specific Comments*

*1) P15, P17 and other places in the manuscript: "On the other hand, the changes in the OMZ intensity have the potential - via denitrification - to alter the marine nitrogen budget, and hence the efficiency of the biological pump of carbon and climate, on the longer timescales." "Therefore, the enhanced denitrification in the Arabian Sea has the potential to significantly reduce biological productivity at the basin scale (and beyond) on timescales of decades to centuries."*

*We usually consider that on long time scales, denitrification and nitrogen fixation com-pensate each other at the global scale. Water masses where denitrification occurs at depth present an excess in available phosphate. When this excess in phosphate makes it back to the surface it can support nitrogen fixation. Could you please dis- cuss your result in this context? On what temporal and spatial scales is your result pertinent? Do you expect a global compensation of this increase in denitrification on longer timescales? How would this impact your conclusion on biological productivity, locally and globally? You briefly discuss the limitation of not having nitrogen fixation in your model but my comment here is more general and calls for some discussion and perspectives on how your results fit in the more global climate change context.*

We speculate that the potential perturbation of the nitrogen (N) and carbon (C) cycles would subside and weaken on timescales that approach the turnover time of fixed nitrogen (2000-3000 years). This is because recent observations and studies suggest a balanced nitrogen budget on the timescales of glacial-interglacial variations (Gruber 2004), thus suggesting a tight coupling between denitrification and $N_2$ fixation on timescales of thousand years (Gruber 2008, Sigman and Haug, 2003). Two negative feedbacks may indeed limit, and eventually reverse, the growth of such denitrification induced perturbation of the N cycle (Deutsch et al, 2004, Gruber 2004). The first feedback is based on the fact that enhanced denitrification, by reducing the inventory of fixed N, would ultimately reduce productivity, and hence export fluxes and $O_2$ demand, which would result in a weakening of the intensity of OMZ and denitrification. The 2$^{nd}$ feedback builds on $N_2$ fixation and the assumption that diazotrophic organisms can outcompete normal phytoplankton in situations of severe fixed N deficits. Hence, enhanced denitrification by favoring the excess of phosphate over nitrate, would favor $N_2$ fixers, and hence would lead to enhanced $N_2$ fixation that would ultimately lead to compensating the initial perturbation and thus restoring the original balance (Gruber 2008). However, there remain large uncertainties regarding the amplitude of these feedbacks and on what timescale they may operate as other factors besides the $NO_3$ to $PO_4$ ratio can control $N_2$ fixation. An example of this is iron availability as $N_2$ fixers have a high iron demand (Falkowski 1997). Furthermore, observations of excess phosphate over nitrate indicate basin-scale decoupling between $N_2$ fixation-dominated regions (e.g., North Atlantic) and denitrification dominated zones (e.g. Arabian Sea), thus suggesting a possible occurrence of important imbalances in the N budgets on timescales shorter than the timescale of the overturning circulation. This is also supported by previous paleoceanographic studies that have shown considerable changes in the past in the N cycle as evidenced by atmospheric $N_2O$ variations during the glacial-interglacial transitions (Fluckiger et al, 1999) as well as large past fluctuations in denitrification (Altabet et al, 1995, 2002).

In response to the reviewer comment, we have expanded our discussion of the potential effects of changes in denitrification on the nitrogen cycle on longer timescales by adding a detailed statement in section 4.2.2 that summarizes the key points exposed in the discussion above (please see the added statement highlighted in red from line 5, p18 to line 9, p19 of the revised manuscript).

*2) In P18, you discuss the production of N2O. Based on previous work on O2 and N2O production, could you compute a first order back of the envelope estimate of how much N2O could be produced by your O2 changes? How does that compare to previous estimates and to the global production of N2O in the ocean and out of the ocean?*

We thank the reviewer for his/her valuable suggestion. In order to address this point, we have reviewed the relevant literature on the sources and sinks of the $N_2O$ in the Arabian Sea and the different parameterizations of the $N_2O$ used in previous modeling studies. Recent $N_2O$ parameterizations (e.g., Martinez-Rey et al., 2015) assume the production of $N_2O$ to result from two major pathways while its consumption occurs in OMZ through denitrification. The first pathway is associated with nitrification (high $O_2$ pathway) and occurs typically at $O_2 >$ 20mmol/m$^3$. The 2$^{nd}$ pathway occurs at low $O_2$ ($< 5$mmol/m$^3$) and involves a combination of nitrification and denitrification (low $O_2$ pathway). The relative contribution of the two pathways is still not well established although recent studies suggest the nitrification pathway to be dominant globally (e.g., Freing et al, 2012). In the Arabian Sea, an observational study by Bange et al, (2001) indicates that $N_2O$ formation via nitrification remains the dominant pathway of $N_2O$ production outside of the OMZ. In the core of the OMZ ($O_2 < 5$mmol/m3), however, data suggests an important production from denitrification combined with $N_2O$ removal near oxygen total depletion (anoxia).

In conclusion, as denitrification leads to both production (under suboxic conditions) and consumption of $N_2O$ (under anoxic conditions), the net effect of a change in denitrification on $N_2O$ total budget is not easy to quantify without a dedicated parameterization of $N_2O$ fully taking into account the different sources and sinks of the nitrous oxide as well as the effect of the transport and gas exchange on its dynamics. Therefore, we could not make any reasonable estimate of the net change in the $N_2O$ that would result from dentrification changes, as this would likely be very sensitive to slight changes in $O_2$ concentrations as well as to the detail of the $N_2O$ parameterization. However, given the fact that the nitrification pathway appears to dominate $N_2O$ production in the AS and since nitrification is predicted to increase by up to 62% in response to a 50% increase in wind stress, we expect the $N_2O$ production to most likely increase in response to monsoon wind intensification.

In response to the reviewer comment, we have added a short paragraph in section 4.2.2 where we discuss the potential changes in $N_2O$ production and consumption terms following the key arguments detailed above. More specifically, we have added the following statement:

*"The increase in the Arabian Sea denitrification should also lead to an increase in the $N_2O$ production. This could not be tested in the present study, as $N_2O$ is not represented in our model. Indeed, as denitrification leads to both production (under suboxic conditions) and consumption (under anoxic conditions) of $N_2O$, the net effect of a change in denitrification on $N_2O$ total budget is not easy to quantify without a dedicated parameterization of $N_2O$ fully taking into account the different sources and sinks of the nitrous oxide as well as the effect of the transport and gas exchange on its dynamics. However, we speculate that significant monsoon intensification has the potential to lead to an important enhancement of $N_2O$*

*production because of enhanced nitrification. Indeed, nitrification is predicted to increase by up to 62% in response to a 50% increase in wind stress while Arabian Sea data suggests nitrification to be the dominant pathway of $N_2O$ production outside of the OMZ and a major contributor, together with denitrification, to its production inside the OMZ, (Bange et al, 2001).}"* (please see lines 10-19, page 19 of the revised manuscript).

*3) P19: "Here we show that the changes in the SW monsoon winds dominate the response of the Arabian Sea ecosystem and that the changes in the NE monsoon play a relatively smaller role. Therefore, our results validate previous paleo studies that assign the dominant role of OMZ oscillations control to the Indian SW summer monsoon (e.g. Schulz et al., 1998; Altabet et al., 2002)."*

*You should discuss why the dominance of the SWM is to be expected: 1) the biological production during the SWM dominates the total annual production and 2) in your model NEM winds primarily increase MLD, ventilation and provides O2 to the region, as shown by the higher increase in the suboxic volume in your SWM+/NEM- simulation than in your SWM+/NEM+ simulation (Fig 5).*

We thank the reviewer for this important comment. We identified three mechanisms that can explain the strong control of the SW monsoon perturbation over the OMZ annual mean response. First, as suggested by the reviewer the biological production during the SW monsoon dominates the annual production (explains more than 40% of the annual levels while NEM productivity contributes by less than 33%) and hence is responsible for a substantial fraction of the annual oxygen consumption at depth. Furthermore, summer productivity is more sensitive to wind changes as it is directly driven by wind-induced upwelling. In contrast, NE monsoon productivity is driven by wintertime convection. Hence, NE monsoon wind intensification enhances vertical mixing and surface nutrient concentrations, but also deepens the mixed layer, thus potentially increasing light limitation. This results in a more limited increase in winter productivity (+38% increase in response to 50% increase in wind stress) in comparison to summer productivity (+52% increase in response to 50% increase in wind stress), thus leading to a weaker increase in $O_2$ consumption during the NE monsoon in comparison to the SW monsoon. Finally, the deepening of the wintertime MLD (up to 25m) that result from NE monsoon intensification enhances the ventilation of the northern and northeastern Arabian Sea, thus compensating the mild increase in $O_2$ consumption that result from enhanced winter productivity.

Following the reviewer's suggestion, we have added the following two statements in section 3.1 and 3.2 to explain the strong control of the SWM perturbation over the NPP and OMZ annual mean responses, respectively:

In section 3.1, we have added:

*"Two factors explain the strong control of the SWM perturbation over the NPP annual mean response. First, the biological production during the SWM dominates the annual production (explains more than 40% of the annual levels while NEM productivity contributes by less than 33%). Second, summer productivity is more*

*sensitive to wind changes as it is directly driven by wind-induced upwelling. In contrast, NEM productivity is driven by wintertime cooling and convection. Hence, NEM wind intensification enhances vertical mixing and surface nutrient concentrations, but also deepens the mixed layer, thus potentially increasing light limitation. This results in a more limited increase in winter productivity (+38% increase in response to 50% increase in wind stress) in comparison to summer productivity (+52% increase in response to 50% increase in wind stress)."* (please see 3.1, pages 11-12 of the revised manuscript).

In section 3.2, we have added:

*"This can be partially explained by the larger summer productivity and its larger sensitivity to wind changes leading to stronger perturbation of the O2 demand. Additionally, the deepening (by up to 25m) of the wintertime mixed layer that result from NEM intensification enhances the ventilation of the northern and northeastern Arabian Sea, thus compensating the mild increase in O2 consumption that result from enhanced winter productivity"*. (please see 3.2, page 12 of the revised manuscript).

*Technical Corrections*
*Figure 4: could you make the numbers on panel b more visible.*

Done.

**Anonymous Referee #2**

*The authors used a ROMS, which was coupled to an NPZD model to study impacts of changing monsoon winds on the OMZ and the marine nitrogen cycle in the Arabian Sea. The results indicate that changes in the summer monsoon winds exert the main control on productivity, the OMZ and finally the marine nitrogen cycle. Intensification of the summer monsoon winds increases the productivity, expands the OMZ at depth, and increases denitrification, while an enhanced intrusion of oxygen-enriched surface water weakens the intensity of the upper OMZ at water-depth between 100 and 200 m. Since there are indications that the Indian summer monsoon intensifies in response to global warming, the topic addressed within the manuscript is of great relevance. The manuscript is, moreover, well-written. However, the presented model results and parameterizations of important processes deviate from conclusions drawn from field data. This in addition to some other aspects needs clarification before publication of the manuscript can be recommended.*

We are thankful to the reviewer #2 for the time spent on reviewing our manuscript and for his/her valuable comments that have made the manuscript stronger. Following the reviewer suggestion, we have added model comparisons with field data to the revised the manuscript to further support our main results. Moreover,

we have added a couple of clarifications as requested by the reviewer. Please see below our responses to specific comments.

*1) As stated in the abstract the main conclusion is as follows: 'We show that the Arabian Sea productivity increases and its OMZ expands and deepens in response to monsoon wind intensification. These responses are dominated by the perturbation of the summer monsoon wind, whereas the changes in the winter monsoon wind play a secondary role'. Here it should be mentioned explicitly that winds are generally weak and winter cooling drives productivity during the winter monsoon (e.g. Madhupratap et al. 1996). In its present form it is misleading because it could imply that wind mixing is a dominant factor because it was selected to run the sensitivity experiment.*

We agree with the reviewer that winter winds are generally weak and that winter cooling and convection drives winter bloom. We have made this more explicit in the revised manuscript by adding the following statement: *"In contrast, NEM productivity is driven by wintertime cooling and convection."* (see page 11, line 11 of the revised manuscript). Additionally, we now better explain the mechanisms through which the perturbation of the summer monsoon controls the OMZ annual mean response (please also see our response to comment #3 by reviewer#1).

*This assumption would furthermore suggest that model results show that the summer monsoon is more important for the productivity as the winter monsoon. The discussion of various pale- oceanographic studies shows that warming increases wind speeds, expands the OMZ and increases denitrification. This, furthermore supports the impression that the summer monsoon is the main driver, and the winter monsoon of lower importance. This was not studied in the model and it should also be considered that these paleoceanographic results were obtained by comparing glacial and interglacial periods. During the Holocene a weakening of the summer monsoon strength seems to be accompanied by an intensification the OMZ (see e.g. Rixen et al. 2014) suggesting that ventilation plays a more important role than implied by the model output*

We have improved our discussion of the mechanisms that lead to stronger control by the summer monsoon perturbation (please see our response to comment #3 by reviewer#1). We would also like to point out that our finding that the summer monsoon driven productivity exceeds that of the winter monsoon is also supported by several observations (e.g., Dickson et al, 2001). Otherwise, we agree that past ventilation changes may have played an important role in modulating the variations in the Arabian Sea OMZ and denitrification as suggested by some previous studies (Pichevin et al, 2007, Boning & Bard, 2009) and already acknowledged in our manuscript (see section 4.2.3 of the manuscript). While our study highlights the strong link between monsoon variations and OMZ fluctuations, it does not rule out a potential contribution from changes in large-scale ventilation. With our current model setup we cannot however test such a hypothesis as this would require using global simulations with a realistic representation of the Arabian Sea OMZ as stated in the manuscript (see section 4.2.3, lines 15-17, page 20).

*2) The occurrence of the secondary nitrite maximum is generally assumed to indicate active denitrification in the water column of the Arabian Sea (see Naqvi et al. 1991, 1998 and more*

*recently Bulow et al., 2010, Gaye et al. 2013). The secondary nitrite maximum occurs a water depth between 100 and 400 m which implies that denitrification is absence or at least of minor importance in the deeper part of the OMZ. The model results show exactly the opposite as summarized in the abstract: 'The increased productivity and deepening of the OMZ also lead to a strong intensification of denitrification at depth, resulting in a substantial amplification of fixed nitrogen depletion in the Arabian Sea'. This needs to be clarified as well as the ignored N-fixation as pointed out by reviewer #1.*

We do not agree with the reviewer statement in that our modeled denitrification profile disagrees with observations. Indeed, our control simulation also shows maximum denitrification between 100 and 400m (black curve in Fig. 7b of the manuscript). For example the rate of simulated denitrification in the control run below 400m is at least a factor 7 smaller than at 200m. Our results are therefore consistent with observations made by studies cited by the reviewer (e.g., Bulow et al., 2010, Gaye et al. 2013). The deepening of denitrification referred to in the statement cited by the reviewer concerns the 50% increased wind perturbation simulation. We do not expect the model subjected to such a relatively extreme perturbation to stay close to observations made under present day forcing.

*3) The parameterization of the carbon export into the deep sea should be described in more detail. Since sinking speeds and respiration rates are provided I assume that a model similar to those introduced by Banse (1990) was used. The considered sinking speeds of 1 and 10 m per day are an order of magnitude lower as those derived from sediment trap studies (see e.g. Berelson, 2001). Please clarify.*

The detail of the representation of the carbon export in the model is given in section 2.1, lines 18-22, page 4. Following the lead of Gruber et al (2006), particle sinking is represented explicitly using 2 detritus classes that can also be advected laterally: a class of large and fast sinking particles (10m d$^{-1}$) and another class of small and slow sinking particles (1m d$^{-1}$). We also specify the remineralization rates used for the large (0.01 d$^{-1}$) and small detritus (0.03 d$^{-1}$). We do not use representations of export based on the Martin equation where the particle flux is set to decrease exponentially with depth such the ones referred to by the reviewer (and described in Banse 1990 or Berelson, 2001). Instead, the flux attenuation with depth emerges from the decomposition of organic matter as it sinks.

We would like to point out that it is the ratio of sinking speed to decomposition rate (corresponding to a remineralization lengthscale) that controls the attenuation of export fluxes in our model. While sinking speeds used in the model can be lower than some sediment trap estimates by up to one order of magnitude as correctly mentioned by the reviewer, the decomposition rates used in the model are also proportionally weaker than in those studies (e.g., ~ 0.2 to 0.3 d$^{-1}$ in Banse 1990, Deep Sea Research). Therefore, despite differences in sinking speed and decomposition rates, the remineralization lengthscales in our model (1000m and 33m for large and small detritus, respectively) are comparable to those implied in

some previous studies. This is supported by the reasonable agreement of our simulated export fluxes with the sediment trap observations from the US JGOFS Arabian Sea expedition (see the new Figure A6 in appendix A of the revised manuscript and our response to comment #4 below).

*4) Among others satellite-derived chlorophyll concentrations were used to validate the model, which to my understanding do not agree well to model outputs. (The months given in Fig. 2 bottom need to be corrected). However, satellite data especially during the summer monsoon are problematic but there are a number of sediment trap data from the Arabian Sea (see e.g. Honjo et al. 1997 and Lee et al. 1997). Considering the importance of carbon export model data should be compared to sediment trap data to make the main conclusions convincing.*

We agree with the reviewer that the modeled chlorophyll-a does not agree well with the observations in certain areas, especially off the coast of Somalia as already acknowledged in the submitted manuscript (lines 9-10, page 8). However, the fidelity of the model north of 10ºN is in line (if not better) with most of state of the art models (e.g., Resplandy et al, 2012). This is also supported by the relatively high correlations and comparable variances between simulated and observed surface chlorophyll-a distributions evidenced in the Taylor diagrams (Fig 4).

Yet, we do agree with reviewer #2 comment that satellite chlorophyll data is not enough to evaluate the biological model and we thank him/her for his/her suggestion to include more field data in the model validation. Following the reviewer suggestion, we invested time to enhance our model evaluation by adding comparisons with field data from the US JGOFS Arabian Sea Process Study (1995). This consists in: i) $^{14}$C primary productivity ii) export fluxes at 100m estimated using $^{234}$Th removal rates and iii) export fluxes estimated from sediment trap data at 500m above the seafloor to avoid including resuspension fluxes as advised in previous works (e.g., Gardener 1992). Fluxes were measured essentially during the year 1995 at 5 sites (M1, M2, M3, M4 and M5) along a transect extending from the Coast of Oman to the central Arabian Sea. Because of the relatively limited number of individual in-situ observations of biological productivity available in this dataset (only 5 measurements at each site), we also used satellite-based productivity estimates obtained using two different algorithms: the Vertically Generalized Production Model (VGPM) (Behrenfeld and Falkowski, 1997a) and the Carbon Based Production Model (CBPM) (Westberry et al., 2008) using data from two sensors (SeaWiFS and MODIS). The results of these comparisons are presented in Fig A5 and Fig A6 shown in the appendix of the revised manuscript.

This comparison shows that the model correctly simulates a decrease in productivity and export fluxes as the distance to the coast increases (Fig A5 and Fig A6). The model, however, substantially underestimates the measured primary productivity in all 5 stations. Some of this mismatch may be due to the fact that the in-situ productivity estimates are all coming from one individual year (1995) and based on only 5 independent measurements at each site (Lee et al, 1998). Given the importance of both mesoscale and interannual variability, the in-situ estimates may therefore not be representative of the long-term climatological conditions simulated by the model. Indeed, a better agreement is obtained between the modeled productivity and estimates based on satellite observations that have a

more extensive temporal coverage (Fig A5). We further contrasted the simulated export fluxes at 100m to estimates from Lee et al (1998) at the 5 stations (Fig A6). Our modeled export fluxes generally overestimate the $^{234}$Th-based estimates but remain comparable in magnitude with these observations. Furthermore, the model reproduces quite accurately the observed offshore gradient in export. It is worth highlighting however that similarly to in-situ measured productivity, these export fluxes are based on 4 independent measurements at each site only, all from the same year. This may induce biases in these estimates due to contamination by mesoscale and interannual variability. We finally compared the modeled export fluxes in the deep ocean (500m above the seafloor) to sediment trap data at the same 5 sites (Fig A6). The comparison shows a good agreement between the model and the observations at all stations. It is worth noting that these deep export flux estimates can be considered as more robust than those at 100m as they are based on a larger number of independent measurements (20-40 measurements at each site).

In conclusion, despite some discrepancies, our modeled fluxes show a reasonable agreement with both field data and satellite observations. Following reviewer's suggestion, we have included a description of these new comparisons in section 2.3 of the revised manuscript (lines 7-28, page 9). We have also corrected the typo in the name of summer months in Fig 2 pointed out by the reviewer.

*5) Considering the overall importance of the selected topic, which will probably attract a wider readership, I recommend to avoid Taylor diagrams and use simple xy scatter plots. They are clear and easy to interpret. Please include also data from the deeper part of the OMZ in the data / model comparison.*

We prefer the Taylor diagrams over xy plots because the former provide a more quantitative and condensed synthesis of model skill. As each dot on the Taylor diagrams represents an independent comparison between the model and the data, replacing the two diagrams with xy plots would require 18 figures! Additionally, because of the large number of individual observations used in this comparison (resulting from the high-resolution of the satellite products and the large number of observations available in World Ocean Database), the xy plots may be visually difficult to read and compare. Therefore, we decided to keep the Taylor diagrams as they are widely used for model evaluation and model skill assessments in climate and environmental sciences. Please also note that Fig 4b include observations sampled down to 1000m (grey filled circles).

*6) Moel et al. 2009 is missing in the reference list.*

Corrected. Thank you!

---

## Author Response (AR2)

Response to the reviewer

*The authors would like to thank the reviewer for his/her feedback and the time he/she took to review our manuscript. Please find below point-by-point replies to the reviewer's comments. Reviewer comments are highlighted in black and author responses are in blue.*

The authors improved the ms but the response of the OMZ to climate change is a crucial issue and involves a complex interaction of processes which are only partly addressed in the sensitivity experiments carried out by Lachkar et al.. As stated by the authors in chapter 4.4.2 first sentence, the study focuses on the sensitivity of OMZ to monsoon winds. That is interesting and I would suggest to adapt the abstract, introduction and conclusion to the focus of the ms.

The referee suggests to "*adapt the abstract, introduction and conclusion to the focus of the manuscript*" which is the study of *"the sensitivity of OMZ to monsoon winds"*. We believe that this is already the case as can be seen in the following statements:

In the abstract:
(lines 5-7)
"*…Yet, the response of the OMZ to these wind changes remains poorly understood and its amplitude and timescale unexplored. Here, we investigate the impacts of perturbations in Indian monsoon wind intensity (from -50% to +50%) on the size and intensity of the Arabian Sea OMZ, and examine the biogeochemical and ecological implications of these changes…*"

(lines 18-20)
"*We conclude that changes in the Indian monsoon can affect, on longer timescales, the large-scale biogeochemical cycles of nitrogen and carbon, with a positive feedback on climate change in the case of stronger winds.*"

In the introduction (p3, lines 31-34):
"*…Here we address these questions and explore the mechanisms by which the Arabian Sea ecosystem responds to monsoon wind changes using a regional eddy-resolving model. We examine how idealized changes in summer and winter monsoon wind intensity affect the productivity and the volumes of hypoxic and suboxic water in the Arabian Sea and explore the biogeochemical and ecological implications of these changes….*"

In the method section (section 2.2, p5, 19-22):
"*Although these runs explore different wind perturbation scenarios, they are highly idealized by nature and are not intended to mimic future projections or realistic future trajectories, but rather aim at exploring the sensitivity of the Arabian Sea OMZ to monsoon wind intensity changes and improving our understanding of the key mechanisms that control the OMZ response and its timescales.*"

In the results section (section 3, p10, 26-27):
"*To explore the sensitivity of the Arabian Sea ecosystem to changes in the intensity of monsoon winds, we consider various scenarios of idealized wind perturbations.*"

In the discussion section (section 4.4.2, p21, lines 30-31, p22, line 1):
*"The primary focus of this study is the sensitivity of the Arabian Sea OMZ to monsoon wind changes and its response timescale. This justifies the use of highly idealized wind perturbations, as our simulations are not intended to mimic realistic future changes but rather to deepen our understanding of the key mechanisms at work and their potential implications."*

In the conclusions (section 5, p23, lines 5-6):
*"A set of coupled physical biogeochemical simulations of the Arabian Sea ecosystem reveals a tight coupling between the intensity of the summer monsoon wind and the size and intensity of the Arabian Sea OMZ".*

At no point in the manuscript we claim the paper addresses the question of the response of the OMZ to climate change which is a much more complex problem that involves perturbations that are not covered in the study (e.g., increased warming and stratification, changes in large scale ventilation, changes in biological productivity unrelated to changes in the winds (for example due to atmospheric deposition of nutrients, etc…)).

Considering that Lachkar et al. are interested in the contemporary and even future ocean in addition to glacial also results obtained from Holocene records should be included in the discussion. First of all because of the similar boundary conditions and secondly also because results from the Holocene record contradict the presented interpretation of the model results.

Our study explores how the Arabian Sea OMZ responds to changes in monsoon winds and the timescales and the mechanisms that drive this change. It is not about reproducing the past evolution of the OMZ or its future trajectory. This is made explicit in p5, line 20 of the revised manuscript, see our statement: "...they are highly idealized by nature and are not intended to mimic past conditions from paleoclimatic reconstructions or future trajectories" (see also p21, line 31).
In order to simulate the conditions of early Holocene and explain the trends the reviewer is referring to, one would need to use realistic atmospheric conditions (not only for winds, but also heat and freshwater fluxes) and lateral boundary conditions of temperature, salinity, nutrients and oxygen that prevailed during this period, as well as a realistic representation of the Red Sea and the Gulf at that time (probably partially or entirely closed because of the much lower sea level). This is an entirely different question that lies far beyond the scope of this study.
However, following the suggestion of the referee we have added to section 4.2.3 a brief discussion of the OMZ changes during the Holocene and how to reconcile our results with the recent findings by Gaye et al. (submitted to Biogeosciences discussions) and Das et al. (2017). Please see our detailed response to next referee's comment and p20, lines 7-12 in the revised manuscript.

For example, looking from today back into the past (the last 10000 years) decreasing d15N values within sedimentary records (especially from center of the OMZ in the north and east) correspond with decreasing burial rates of organic matter and an increasing summer monsoon strength. This lead to the conclusion that a strengthening of the summer monsoon weakens the OMZ due to associated changes in the ocean's circulation and ventilation during the Holocene. Despite critics of the authors on global models recent results obtained from a global ocean circulation model support this conclusion by showing an increasing volume of oxygen depleted water (OMZ) associated with nearly constant productivity and weakening of the summer monsoon during the Holocene (from the past to the present).

See Gaye et al. 2017 https://www.biogeosciences-discuss.net/bg-2017-256/. and older references therein.

We do not believe there is a contradiction between our results and the findings of the study the reviewer is referring to. Our study confirms the strong link that exists between the strength of SW monsoon winds and the intensity of the Arabian Sea OMZ and denitrification levels in agreement with a large number of paleoceanographic studies that suggest higher OMZ intensity and elevated denitrification rates during warm periods (with stronger SW monsoon) and weaker OMZ and reduced or absent denitrification during cold periods (Altabet 1995, Altabet 1999, 2002, Reichart et al 1998, Schulte et al. 1999). However, we do not claim that the Arabian Sea OMZ intensity is completely determined by the intensity of the monsoon winds. Instead, our study shows that if only monsoon winds are perturbed (with the assumption that everything else is kept constant, which seems not to be the case during the late Holocene), then the OMZ and denitrification will strongly increase in response to monsoon intensification. We also recognize among the study caveats the fact that "we considered the effects of monsoon changes in isolation" (see p23, lines 1-3).

The fact that an intensification of the OMZ may have coincided at certain point in time (middle to late Holocene: ~4200 years BP onwards according to Das et al 2017) with a weakening of the SW monsoon does not lead to the conclusion that a weakening of the monsoon causes an intensification of the OMZ (or a strengthening of the summer monsoon weakens the OMZ as put by the referee). Instead, this means that the OMZ intensity is not entirely driven by the SW monsoon intensity, but can be affected by other factors (such as changes in large-scale ventilation and fluctuations in exchange with Red Sea and Arabian/Persian Gulf) as suggested by some previous studies (e.g., Pichevin et al, 2007, Boning and Bard 2009, Das et al, 2017) and already acknowledged in our manuscript (see p20, lines 1-4).

We note that the response timescale to monsoon wind changes can be relatively short (decades to centuries) while OMZ fluctuations involving large-scale ventilation changes are associated with much longer timescales (centuries to thousands of years). Therefore, this suggests that abrupt changes in denitrification and OMZ intensity recorded in marine sediment data (e.g., during Dansgaard-Oeschger events or Heinrich events) are more likely to result from monsoon wind changes than changes in ventilation by large-scale circulation or marginal seas.

To avoid any misinterpretation of our results, we further stress the potential role of ventilation changes in the discussion section (4.2.3). More specifically, we state: *"Besides monsoon strength, some studies have linked past OMZ intensity changes to changes in the rate of formation and subduction of oxygen enriched Subantarctic Mode Water (SAMW) and Antarctic Intermediate Water (AAIW) in the Southern Ocean in association with Atlantic Meridional Overturning Circulation (AMOC) fluctuations (Pichevin et al, 2007; Boning and Bard, 2009)."* (please see p20, lines 1-4 in the revised manuscript).

Regarding the Holocene, we have added: *"In the Holocene, large-scale ventilation changes may have played an important role together with the fluctuations in monsoon intensity as suggested by recent studies (Das et al, 2017, Gaye et al., submitted to Biogeosciences discussions). For instance, recent paleo reconstructions by Das et al. (2017) suggest an intensification of the Arabian Sea OMZ from the middle to late Holocene despite a weakening of the SW monsoon winds. These authors hypothesize that the recent (~ 4000 years BP onwards) decline in oxygen at depth may have resulted from a cut off of the Arabian Sea OMZ from the oxygen enriched AAIW and SAMW."* (Please see p20, lines 7-12 in the revised manuscript).

Furthermore the authors are correct figure 7b shows in line with observations the denitrification peak immediately below the mixed layer. Denitrification at depth between 400 and 1300 m with a small secondary peak at 500 m is unsupported by observations. This means there are two peaks: one shallow one supported by observations and another deep one which is unsupported by observations.

We would like to highlight that what is shown in Fig7b is domain integrated denitrification (in Gmol per meter of depth) as a function of depth. This cannot be compared to individual profiles from observations. Indeed, because of the very limited number of available direct measurements of denitrification rates, especially at depths below 300m, no equivalent can be derived from observations. This is particularly true because of the patchiness of denitrification in space and time that results in strong heterogeneity in denitrification profiles.

Our domain-integrated denitrification is maximum between 100 and 300m, but shows a much (a factor 6 or 7) weaker secondary maximum between 500 and 800m. This secondary maximum can be explained by the fact that the (potentially denitrifiying) suboxic area is largest at this depth (see Fig7a) and denitrification is still detectable in our model at this depth, although occurring at more than an order of magnitude smaller rate (on per unit of volume basis) than in the upper 200m.

We searched the relevant literature and found no evidence pointing towards a complete shutdown of denitrification below 400m as can be inferred from the reviewer's comment. Actual measurements of denitrification rates (incubation) have been few, but in these few studies, denitrification was found at all sampled depths (down to 350m for Devol et al, 2006 and down to 400m for Bulow et al (2010). In Devol et al (2006), there was no clear depth structure found between 150 and 300m. In Bulow et al (2010), denitrification rates were largest in the layer between 150-200m, but significant denitrification was found down to 400m. No samples were taken below this depth (400m) in any of the two studies.

Most previous estimates of denitrification in the Arabian Sea are based on indirect methods involving the concept of nitrate deficit (a pool of missing nitrate resulting from denitrification) that involves stoichiometric ratios between nitrate, phosphate and oxygen together with estimates of residence times of denitrifying water masses. An alternative way to express nitrogen anomalies is through nitrogen gas excess. The large uncertainties associated with these assumptions result in large uncertainty in the derived denitrification estimates that vary by more than a factor four between the different studies (Naqvi, 1987 ; Mantoura et al., 1993; Howell et al., 1997 ; Codispoti et al., 2001).

Several previous studies using indirect methods suggest the occurrence of occasional and weak (but still detectable and non negligible) denitrification well below 400m. For instance, evidence from Morrison et al (1999) shows important variability in vertical profiles of nitrate deficits estimated at 4 JGOFS stations in the Arabian Sea in 1995 with significant nitrate deficits reaching as deep as 1000m in one station. Naqvi (1994) shows significant nitrate deficits down to 1000m. He further notes: "a deeper (700-1200m) denitrifying layer may also develop occasionally, probably due to advection of nepheloids layers from the continental margins".
Bange et al. (2001) has linked the secondary peak in $N_2O$ between 800-1000m in the Arabian Sea to occasional denitrification. In Codispoti et al (2001) and Devol et al, (2006), both nitrate deficit and nitrogen excess (indicative of denitrification) are found to be maximum around 200-400m, but remain significant down to 1000-1500m and below.

Because nitrate deficits (and nitrogen excess) reflect denitrification integrated over time, accumulation of nitrite, an intermediate in the process of denitrification, has conventionally been used as an indicator of active denitrification. Most available nitrite profiles in the Arabian Sea show high concentrations between 200 and 400m (secondary nitrate maximum: SNM) and very low concentrations below.
Yet, Naqvi (1994) shows significant nitrite concentrations reaching down to 500m in the northern Arabian Sea. In Morrison et al (1999) the SNM is shown to reach as deep as 600m. In another study by Brand and Griffiths (2008), measurements of nitrite from a series of cruises between March and October 2003 in the north east Arabian Sea shows significant nitrite concentrations down to 1000m with a maximum at around 600m. Finally, Sokoll et al (2012) measured significant nitrite concentrations down to 600m in the northeast Arabian Sea off the coast of Pakistan.

Finally, a recent study by Lam et al. (2011) shows a decoupling between nitrite concentrations and measured denitrification rates between the Omani shelf region and the central Arabian Sea. These authors show that the accumulation of nitrite results from a production of nitrite (through denitrification) that exceeds its consumption rate (due to denitrification and anamox) and conclude that the absence of nitrite (or its presence at very small concentrations) does not necessarily imply an absence of denitrification. This questions the traditional use of nitrite as an indicator of active denitrification.

In summary, the double peak structure that characterizes the depth profile of the domain-integrated denitrification in our model can be explained by: 1) the very high denitrification rates in the top 200m and 2) the presence of weak denitrification at

depth across a very wide area occupied by suboxic waters (500-1000m). Our model results are not conflicting with data based evidence as suggested by referee, as the presence of weak denitrification at depth (>400m) has been reported in several previous studies as shown earlier. Furthermore, the total water column denitrification in our model (18.5 Tg N/yr) lies within the range of previously published data-based estimates that range from 10 to 44 TgN/yr (Naqvi, 1987 ; Mantoura et al., 1993; Howell et al., 1997 ; Codispoti et al., 2001).

For more clarity, we added in the revised manuscript a statement that explains the secondary maximum in integrated denitrification profile shown at depths between 500 and 800m. Please see caption of Fig. 7 where we have added the following text: *"Note the presence of a weak secondary maximum in domain-integrated denitrification between 500 and 800m in all simulations. In the control simulation, this can be explained by the fact that at this depth range the area occupied by (potentially denitrifying) suboxic water is largest and weak denitrification is still present although at an order of magnitude weaker rate (on per unit of volume basis) than at 200 m."*

By increasing wind speeds the supported peak decreased and the unsupported peak increased. Since the latter overcompensated the first the authors concluded an intensifications of the summer monsoon winds deepens the zone of denitrification and increase denitrification rates. To base a conclusion on the intensification of an unsupported peak is to my understanding questionable. Ignoring the response of the unsupported peak would show that stronger summer monsoon winds lower denitrification, which is in line with the Holocene d15N records and the global ocean circulation model. These aspects should also be considered in the discussion and conclusion.

The increase of denitrification at depth and its weakening near the surface is a direct consequence of the expansion of the suboxic area at depth and its shrinking near the surface (see Fig7a). This is a consequence of enhanced upper (0-200m) ocean ventilation and increased biological consumption at depth (depth >200m) as evidenced by the oxygen budget presented in Figure 9. We cannot "ignore" the deep OMZ response to make the results follow a scenario likely involving very different boundary conditions (e.g., potential changes in the oxygen lateral boundary conditions during the Holocene).

Some more minor comment refer to the discussion about sinking speeds and respirations rates, light limitation, classification of zones. If it does not matter whether one uses high sinking speeds and respirations rates or low sinking speed and low respiration rates why do not stick to observation in order to avoid such discussions.

The values of sinking speed and remineralization rates used in the model are the default values used in the Gruber et al., (2006) NPZD model.

Please explain in more detail how light limitation was identified and translated into reduced carbon export rates and the why unused nutrient were not consumed somewhere else?

Our reasoning refers to Sverdrup's critical depth (SCD) hypothesis and the observation that winter phytoplankton growth can be limited by the amount of available light resources. A deepening of the mixed layer associated with an increase in upper ocean turbulence moves rapidly phytoplankton through the mixed layer where average available amount light can decrease if the turbulent mixed layer gets deeper than the euphotic zone (or more precisely the compensation depth defined as the depth where the phytoplankton loss terms such as mortality, grazing and respiration compensate photosynthesis). For a review of SCD see Franks (2015).

Following the reviewer's suggestion we have added a few statements to further clarify how the deepening of the winter mixed layer could increase the light limitation and limit productivity increase. More specifically, we have added the following text to section 3.1 (page 12, lines 5-9): *"Indeed, winter turbulent mixed layer deepens in the northern Arabian Sea by up to 20-25m and penetrates below the euphotic zone (1% light depth) at 65-70m. This increases the average exposure of phytoplankton to light-limited conditions, thus potentially limiting the net growth (i.e., gross photosynthetic rate minus loss terms due to mortality, grazing, sinking and respiration) over the water column, and hence reducing the potential biomass and productivity (Franks 2015)."*

According to fig 7a I would suggest to define upper OMZ from below the mixed layer to approximately 375 m (supported denitrifying zone) and the mesopelagic zone between 375 and 1800 m (unsupported denitrifying zone).

We assume the reviewer is referring to the vertical layers (epipelagic 0-200m, mesopelagic 200-1000m and bathypelagic >1000m) used to highlight the OMZ response timescales (Fig. 8) and $O_2$ budgets (Fig. 9). Our use of the three layers is intended to highlight how the OMZ response timescales vary as a function of depth. The response is fastest in the upper (0-200m) ocean and slowest in the deep ocean (>1000m). Changing slightly the definition of the 3 layers won't change the conclusion, as the increase in the timescale of the OMZ response translates the differences in ventilation (circulation) timescales that exist between the surface, the intermediate and the deep ocean. Furthermore, the chosen layers have direct ecological and biogeochemical implications as the top (0-200m) layer corresponds roughly to habitats of epipelagic fishes while the mesopelagic layer (200-1000m) is the region where 90% of oxygen biological consumption occurs (Robinson 2010). Therefore, we prefer to stick to our definition of the three vertical layers.

References:

Altabet, M. A., Francois, R., Murray, D. W., and Prell, W. L.: Climate-related variations in denitrification in the Arabian Sea from sediment 15N/14N ratios, Nature, 373, 506–509, 1995.

Altabet, M. A., Murray, D.W., and Prell,W. L.: Climatically linked oscillations in Arabian Sea denitrification over the past 1 my: Implications for the marine N cycle, Paleoceanography, 14, 732–743, 1999.

Altabet, M. A., Higginson, M. J., and Murray, D. W.: The effect of millennial-scale changes in Arabian Sea denitrification on atmospheric CO2, Nature, 415, 159–162, 2002.

Bange, H. W., S., Rapsomanikis, and M. O., Andreae: Nitrous oxide cycling in the Arabian Sea, J. Geophys. Res., 106(C1), 1053–1065, doi:10.1029/1999JC000284, 2001.

Böning, P. and Bard, E.: Millennial/centennial-scale thermocline ventilation changes in the Indian Ocean as reflected by aragonite preservation and geochemical variations in Arabian Sea sediments, Geochimica et Cosmochimica Acta, 73, 6771–6788, 2009.

Brand, T. D., and C. Griffiths (2008), Seasonality in the hydrography and biogeochemistry across the Pakistan margin of the NW Arabian Sea, Deep Sea Res., Part II, 56, 283–295.

Bulow, S. E., J. J. Rich, H. S. Naik, A. K. Pratihary, and B. B. Ward (2010), Denitrification exceeds anammox as a nitrogen loss pathway in the Arabian Sea oxygen minimum zone, Deep Sea Res., Part I, 57, 384–393.

Codispoti, L.A., Brandes, J.A., Christensen, J.P., Devol, A.H., Navi, S.W.A., Pearl, H.W., Yoshinari, T.: The oceanic fixed nitrogen and nitrous oxide budgets: Moving targets as we enter the anthropocene? Sc. Mar., 65, 85–105, 2001.

Das, M., Singh,  R. K., Gupta, A. K., Bhaumik, A. K.: Holocene strengthening of the Oxygen Minimum Zone in the northwestern Arabian Sea linked to changes in intermediate water circulation or Indian monsoon intensity?, In Palaeogeography, Palaeoclimatology, Palaeoecology, Volume 483, 2017, Pages 125-135, ISSN 0031-0182, https://doi.org/10.1016/j.palaeo.2016.10.035.

Devol, A. H., Uhlenhopp, A. G., Naqvi, S. W. A., Brandes, J. A., Jayakumar, D. A., Naik, H., Gaurin, S., Codispoti, L. A., and Yoshinari, T.: Denitrification rates and

excess nitrogen gas concentrations in the Arabian Sea oxygen deficient zone, Deep-Sea Res. Pt. I, 53, 1533– 1547, 2006.

Franks, P.: Has Sverdrup's critical depth hypothesis been tested? Mixed layers vs. turbulent layers, ICES Journal of Marine Science, Volume 72, Issue 6, 1 August 2015, Pages 1897–1907, 2015.

Gaye, B., Böll, A., Segschneider, J., Burdanowitz, N., Emeis, K.-C., Ramaswamy, V., Lahajnar, N., Lückge, A., and Rixen, T.: Glacial-Interglacial changes and Holocene variations in Arabian Sea denitrification, Biogeosciences Discuss., https://doi.org/10.5194/bg-2017-256, in review, 2017.

Gruber, N., H. Frenzel, S. C. Doney, P. Marchesiello, J. C. McWilliams, J. R. Moisan, J. J. Oram, G.-K. Plattner, and K. D. Stolzenbach: Eddy-resolving simulation of plankton ecosystem dynamics in the California Current System, Deep Sea Res. I, 53(9), 1483–1516, 2006.

Howell, E.A., Doney, S.C., Fine, R.A., Olson, D.B., 1997. Geochemical estimates of denitrification in the Arabian Sea and Bay of Bengal during WOCE. Geophysical Research Letters 24, 2549.

Lam, P., Jensen, M. M., Kock, A., Lettmann, K. A., Plancherel, Y., Lavik, G., Bange, H. W., and Kuypers, M. M. M.: Origin and fate of the secondary nitrite maximum in the Arabian Sea, Biogeosciences, 8, 1565–1577, doi:10.5194/bg-8-1565-2011, 2011.

Mantoura, R.F.C., Law, C.S., Owens, N.P.J., Burkill, P.H., Woodward, M.S., Howland, R.J.M., Llewellyn, C.A., 1993. Nitrogen biogeochemical cycling in the northwestern Indian Ocean. Deep-Sea Research 40, 651–671.

Morrison, J. M., Codispoti, L. A., Smith, S. L., Wishner, K., Flagg, C., Gardner, W. D., Gaurin, S., Naqvi, S. W. A., Manghnani, V., Prosperie, L., and Gundersen, J. S.: The oxygen minimum zone in the Arabian Sea during 1995, Deep-Sea Res. Pt. II, 46, 1903– 1931, 1999.

Naqvi, S.W.A., 1987. Some aspects of the oxygen-deficient conditions and denitrification in the Arabian Sea. Journal of Marine Research 45, 1049–1072.

Naqvi, S.W.A.: Denitrification processes in the Arabian Sea. In: Lal, D. (Ed.), Biogeochemistry of the Arabian Sea. Indian Academy of Sciences, Bangalore, India, pp. 181-202, 1994.

Pichevin, L., Bard, E., Martinez, P., and Billy, I.: Evidence of ventilation changes in the Arabian Sea during the late Quaternary: Implication for denitrification and nitrous oxide emission, Global Biogeochemical Cycles, 21, 2007.

Reichart, G.J., Lourens, J., Zachariasse, W.J.: Temporal variability in the northern Arabian Sea Oxygen Minimum Zone (OMZ) during the last 225,000 years. Paleoceanography 13, 607–621, 1998.

Robinson, C., Steinberg, D. K., Anderson, T. R., Aristegui, J., Carl- son, C. A., Frost, J. R., Ghiglione, J.-F., Hernandez-Leon, S., Jackson, G. A., Koppelmann, R., Queguiner, B., Ragueneau, O., Rassoulzadegan, F., Robison, B. H., Tambourini, C., Tanaka, T., Wishner, K. F., and Zhang, J.: Mesopelagic zone ecology and biogeochemistry – a synthesis, Deep-Sea Res. Pt. II, 57, 1504– 1518, 2010.

Schulte, S, Rostek, F., Bard, E., Rullkötter, J., Marchal, O.: Variations of oxygen-minimum and primary productivity recorded in sediments of the Arabian Sea. Earth and Planetary Science Letters, 173(3), 205-221, 1999.

Sokoll S, Holtappels M, Lam P, Collins G, Schlüter M, Lavik G and Kuypers MMM: Benthic nitrogen loss in the Arabian Sea off Pakistan. Front. Microbio. 3:395. doi: 10.3389/fmicb.2012.00395, 2012.

---

## Author Response (AR4)

Dear Editor,

Thank you for your report and feedback.
As requested, we provide point-by-point response to the reviewer's comments in the attached document.

Please, note that the role of ventilation changes resulting from monsoon wind perturbations is well taken into account in our simulations (please see our response to reviewer's comment #1).
The study does not consider only the ventilation changes that are independent of monsoon wind intensity (for instance associated with circulation changes at the global or basin scales).

We do recognize however that the concomitant occurrence of such additional perturbations in conjunction with monsoon wind changes may affect the overall response of the OMZ. Thus, we revised the abstract and conclusion to make this clearer as you and the reviewer suggested.

We hope that with these additional clarifications and final revisions you will find our manuscript suitable to publication in Biogeosciences.

Thank you for your time and dedication.

Best regards,

Zouhair Lachkar and co-authors

**Response to the reviewer**

We thank the reviewer for his final feedback. Please find below point-by-point replies to the reviewer's comments. Reviewer comments are highlighted in black and author responses are in blue.

1) Link of the model result to future projections and the past evolution of the OMZ.

Yes, it is clearly stated in the ms that the simulations are highly idealized and are not intended to reproduce the past evolution of the OMZ or to predict its future trajectory and that they rather aim at exploring the sensitivity of the Arabian Sea OMZ to monsoon wind intensity changes but this is not reflected in the abstract and conclusion.

In the conclusion we can read : A set of coupled physical biogeochemical simulations of the Arabian Sea ecosystem reveals a tight coupling between the intensity of the

summer monsoon wind and the size and intensity of the Arabian Sea OMZ. We find that the OMZ and ecosystem responses are largely determined by the perturbation of the summer SW monsoon ….

In the abstract the authors wrote that the model results show that the Arabian Sea productivity increases and its OMZ expands and deepens in response to monsoon wind intensification and that this lead to a strong intensification of denitrification at depth, resulting in a substantial amplification of fixed nitrogen depletion in the Arabian Sea.

This is misleading without mentioning the limitation of their simulation and that ventilation change in response to varying monsoon strength can cause the opposite.

The changes in the ventilation that are driven by monsoon wind changes are already represented in the model and are taken into account. Additionally, their effects on oxygen are quantified as can clearly be seen in the oxygen budget presented in Fig. 9 and Fig A9. While enhanced ventilation opposes the effect of increased biological consumption of oxygen, its effect is smaller in magnitude than the effect of increased productivity, except in the upper 200m. This is discussed in detail in section 4.1.

What we did not cover in the study are the changes in large-scale ventilation (and circulation) that are independent of local wind changes (e.g., changes in the global thermohaline circulation). For more clarity and following the reviewer suggestion, we have added statements in the abstract and the conclusion to further highlight that we are considering the effect of wind changes in isolation and that the overall OMZ response may also depend on changes in large-scale ventilation and stratification.

In the abstract, we have added: "Additional potential changes in large-scale ocean ventilation and stratification may affect the sensitivity of the Arabian Sea OMZ to monsoon intensification." (see lines 20-21 in the revised abstract).

In the conclusion, we have added: "These results are obtained while considering the effects of monsoon wind changes in isolation. The response of the OMZ to wind increase may differ however in the presence of other concomitant perturbations such as potential changes in large-scale circulation and ventilation or additional surface warming." (see lines 30-32, p23).

2) The presented idea that large-scale ventilation changes act on longer and changes in wind intensity on shorter time scales is not supported by sediment tarp results. They indicate that increased winds speeds intensify upwelling but at a certain level decrease carbon export because of associated ventilation changes (Rixen et al 1996). This should be considered in the discussion.

The study by Rixen et al (1996) compares export fluxes at 3000m at three stations in the Arabian Sea with the intensity of SW monsoon winds at the trap locations between 1986 and 1992. There is nothing in this study that concerns the changes in the large-scale (e.g., basin-scale) circulation or ventilation or the OMZ response timescales. Therefore, we do not think this is relevant to our work nor to the discussion of the timescales of large-scale ventilation changes.

3) That in the Holocene, large-scale ventilation changes may have played an important role together with fluctuations in monsoon intensity was not only suggested by Gaye et al and Das at 2017 but already in 2014 by Rixen et al 2014 (Biogeosciences) which should accordingly be cited.

Done. Citation added.

4) However my main critics it that the authors present conclusion drawn form simulations (see abstract and conclusion) without mentioning the limitation of their simulation and that ventilation change in response to varying monsoon strength caused the opposite. This is misleading but can be solved by modifying the abstract and conclusion.

Done. See our response to previous comment #1.

2) First of all I have suggested to discuss not to ignore the deep denitrification peak. That the response of low denitrification rates at depth could be more important as those of at shallower depth for the total denitrification is an interesting result. It suggests to pay more attention to denitrification at greater water-depth in future field studies.

Thanks for the clarification. We agree with the reviewer's conclusion.

5) The third point refers to the comment on sinking speed and respiration rates: That previous studies did it same is not an scientific argument.

As we have shown in our previous response, using the historical parameters of the model resulted in a good agreement with observations. Changing the sinking speed and remineralization rate in such a way that the remineralization depth remains constant will not lead to any change in the results as explained in our previous response to a similar comment.